# The nanoscale organization of the Nipah virus fusion protein informs new membrane fusion mechanisms

Qian Wang[1], Jinxin Liu[1], Yuhang Luo[1], Vicky Kliemke[1], Giuliana Leonarda Matta[1], Jingjing Wang[1], Qian Liu[1,2]*

[1]Institute of Parasitology, Faculty of Agricultural and Environmental Sciences, McGill University, Montreal, Canada; [2]Mark Wainberg Center for Viral Diseases, Lady Davis Institute, Montreal, Canada

## eLife Assessment

This **valuable** study advances our understanding of how Nipah virus fusion protein F (NiV-F) organizes into nanoclusters on cell and viral membranes using biochemical and super-resolution microscopy methods. The conclusions are supported by **solid** evidence and the revision has addressed most of the reviewers' concerns. The relationship between clustering and fusion is of high interest and an interesting hypothesis to continue investigating in future studies.

*For correspondence:
qian.liu3@mcgill.ca

Competing interest: The authors declare that no competing interests exist.

**Abstract** Paramyxovirus membrane fusion requires an attachment protein for receptor binding and a fusion protein for membrane fusion triggering. Nipah virus (NiV) attachment protein (G) binds to ephrinB2 or -B3 receptors, and fusion protein (F) mediates membrane fusion. NiV-F is a class I fusion protein and is activated by endosomal cleavage. The crystal structure of a soluble GCN4-decorated NiV-F shows a hexamer-of-trimer assembly. Here, we used single-molecule localization microscopy to quantify the NiV-F distribution and organization on cell and virus-like particle membranes at a nanometer precision. We found that NiV-F on biological membranes forms distinctive clusters that are independent of endosomal cleavage or expression levels. The sequestration of NiV-F into dense clusters favors membrane fusion triggering. The nano-distribution and organization of NiV-F are susceptible to mutations at the hexamer-of-trimer interface, and the putative oligomerization motif on the transmembrane domain. We also show that NiV-F nanoclusters are maintained by NiV-F–AP-2 interactions and the clathrin coat assembly. We propose that the organization of NiV-F into nanoclusters facilitates membrane fusion triggering by a mixed population of NiV-F molecules with varied degrees of cleavage and opportunities for interacting with the NiV-G/receptor complex. These observations provide insights into the in situ organization and activation mechanisms of the NiV fusion machinery.

## Introduction

The family of *Paramyxoviridae* contains many clinically important viruses, such as human parainfluenza virus 1–3, measles virus, and Nipah virus (NiV). Paramyxoviruses code for two glycoproteins on the viral membranes for virus–cell membrane fusion. The receptor-binding proteins (RBP, HN/H/G) engage the host receptors and activate the refolding of the fusion protein (F) that merges the virus and cell membranes (***Navaratnarajah et al., 2020***). Paramyxovirus-F is a class I fusion protein, along with human immunodeficiency virus (HIV) envelope, influenza virus hemagglutinin, and severe acute respiratory syndrome coronavirus-2 (SARS-CoV-2) spike (***Navaratnarajah et al., 2020***). Paramyxovirus-F

is a trimeric transmembrane protein with a 'tree-like' structure at its prefusion conformation. Upon triggering by the RBP/receptor complex, paramyxovirus-F mediates fusion by inserting its N-terminal fusion peptides into the target membrane and undergoes a conformational change from pre- to post-fusion conformations, similar to other class I fusion proteins. The refolding of the paramyxovirus-F overcomes the energy barrier for membrane fusion and leads to virus–cell and cell–cell membrane fusion.

NiV is a well-studied paramyxovirus due to its capability of human–human and human–animal trans-missions, high mortality, and morbidity rates, and the unavailability of vaccines or therapeutics for human use (*Haas and Lee, 2023*; *Shariff, 2019*). NiV is closely related to Hendra virus, Cedar virus, and the newly identified Langya virus (*Thibault et al., 2017*). NiV-F is expressed on the cell surface as an fusion-inactive form, activated in the endosome by cathepsin B and L cleavage, and exocytosed to the cell surface as a fusion-active form (*Diederich et al., 2012*; *Diederich et al., 2005*). NiV-G binds to ephrinB2 and/or -B3 receptors in host cells and triggers the refolding of F that leads to virus–cell membrane fusion (*Negrete et al., 2005*). Three complementary models of the NiV fusion activation mechanism have been proposed in the past. *In the first model*, the NiV-G and F form a complex before receptor engagement. The F/G interaction maintains F at a prefusion conformation. The receptor binding to G induces conformational changes in G, leading to the release and refolding of F. This model is supported by co-immunoprecipitation and conformational antibody-binding assays (*Liu et al., 2015*; *Liu et al., 2013*). *In the second model*, NiV-F exists as hexamer-of-trimers, and the fusion peptides are sequestered at the hexameric interface. The activation of one single trimer by the NiV-G/ephrinB2 complex can be transmitted to all six trimers via the interfaces of the hexamer-of-trimers, leading to efficient F triggering and fusion pore formation. This model is supported by the crystal structure of a GCN4-decorated NiV-F ectodomain and mutagenesis analysis of key amino acid residues at the hexameric interfaces (*Xu et al., 2015*). *In a third model*, NiV-F and G do not form a complex before receptor binding. The NiV-G/ephrinB2 engagement clusters ephrinB2 and triggers F. F triggering leads to reduced mobility of a portion of F molecules that potentially contributes to fusion pore formation. This model is supported by the nanoscale distribution of NiV-F and G on cell membranes resolved by super-resolution microscopy and a single-particle tracking assay of NiV fusion protein at the interface between supported lipid bilayer and live cell membranes (*Liu et al., 2018*; *Wong et al., 2021*). Although these models explain the NiV membrane fusion mechanisms from different perspectives, a link between the molecular structure and the fusion machinery on biological membranes remains unclear.

We attempted to address this link by analyzing the nano-organization of NiV-F at the prefusion state on biological membranes using single-molecule localization microscopy (SMLM) and mutagenesis analyses. We have shown that NiV-F forms clusters on biological membranes that are isolated from NiV-G (*Liu et al., 2018*). Here, we show that NiV-F forms regular-sized nanoclusters on cell membranes regardless of the expression level or endosomal cleavage. The estimated size of NiV-F clusters on cell membrane is similar to that of the hexamer-of-trimer assemblies formed by the GCN4-decorated soluble NiV-F. The NiV-F nano-organization is altered by mutations at the hexameric inter-face and the putative oligomerization motifs at the transmembrane domain (TMD). NiV-F molecules enriched in nanoclusters favor membrane fusion activation. The NiV-F nanoclusters are stabilized by the interactions between NiV-F, the endocytosis adaptor complex AP-2, and the clathrin coat at the cell membranes. In summary, our study reveals the NiV-F nano-organization on the biological membranes and provides novel insights into the NiV fusion machinery in situ.

## Results
### NiV-F forms regular-sized clusters that are not affected by the surface expression level

We recently published that NiV-F formed distinctive nanoclusters on the plasma membrane regardless of the presence of NiV-G or ephrinB2 (*Liu et al., 2018*). To investigate the determinants of the NiV-F nano-organization, we used a custom-built SMLM to probe the NiV-F at various conditions at a 10-nm precision (*Figure 1—figure supplement 1A*). SMLM generates super-resolution images by localizing spatially far-apart fluorophores with nanometer precision (*Coelho et al., 2020*). PK13 cells have a flat morphology that is suitable for SMLM imaging. PK13 cells express little endogenous ephrinB2 and

-B3 receptors for NiV and thus can eliminate any potential receptor-dependent clustering. Cell surface NiV-F was detected using an FLAG tag in its extracellular domain (*Liu et al., 2018*). The FLAG tags were employed to detect NiV-F because of the availability of high affinity and specificity anti-FLAG antibodies for a high labeling density, which is vital for the reconstruction accuracy of SMLM images (*Dempsey et al., 2011*). To rule out the effect of the FLAG tag on NiV-F clustering, we compared NiV-F clusters formed by NiV-F-FLAG to that of a NiV-F-HA construct. Both HA and FLAG tags were inserted after amino acid residue 104, right before the fusion peptide (*Figure 1—figure supplement 1B*). Visual examination of the SMLM images reveals that NiV-F-FLAG and NiV-F-HA form similar clusters (*Figure 1—figure supplement 1C*). The clustering tendency, indicated by Hopkins' index, is comparable between NiV-F-HA and NiV-F-FLAG (*Figure 1—figure supplement 1D*), suggesting that NiV-F clustering is not affected by specific epitope tags. To test whether the FLAG tag affects the function of NiV-F, we pseudotyped NiV-G-HA and NiV-F-FLAG or untagged NiV-F on the recombinant vesicular stomatitis virus (VSV) with glycoprotein (G) replaced by a *Renilla* luciferase gene (VSV-ΔG-rluc) and determined the viral entry levels (VSV/NiV pseudoviruses). The virus entry induced by NiV-F-FLAG is comparable to that of untagged NiV-F (*Figure 1—figure supplement 1E*), verifying that the FLAG tag has minimal effect on the membrane fusion activity of NiV-F.

Next, we investigated whether the nano-organization of NiV-F depends on the cell surface expression (CSE) levels. The total CSE level of NiV-F in a selected cell was measured before the cell was subjected to SMLM imaging. The SMLM images show that NiV-F forms similar nanoclusters in both high- and low-expression cells (*Figure 1A, B*). Our data show that the clustering tendency of NiV-F is independent of its CSE in individual cells (*Figure 1C*). To gain a quantitative understanding of NiV-F nanoscale organization, we used a cluster identification method DBSCAN (Density-Based Spatial Clustering of Applications with Noise) that links the closely situated localizations in a propagative manner. We created cluster and density maps using Clus-DoC for representative high- and low-expression cells (*Figure 1D, E*; *Pageon et al., 2016*). Our data show that the estimated diameter of the NiV-F clusters ranges from 20 to 34 nm, with a peak at 24 nm (*Figure 1F*). Nonetheless, the precise values of the cluster diameter are indicative rather than definite in SMLM as it could be affected by the labeling complex and the blinking property of the fluorophore. The distribution of the cluster diameter shows that ~60% of NiV-F clusters have similar sizes, indicating that NiV-F clusters maintain a quite uniform appearance. Interestingly, we noticed that the size of the clusters (*Figure 1G*) and localization density (*Figure 1H*) in clusters were similar between high- and low-expression cells, suggesting that the NiV-F organization in nanoclusters is not affected by the expression levels. Moreover, our data show that the high-expression cells have more clusters than the low-expression cells (*Figure 1I*). These results suggest that changes in CSE may affect the number of NiV-F clusters on the plasma membrane (*Figure 1I*), but not the clustering tendency of NiV-F (*Figure 1C*), the size of the clusters (*Figure 1G*), or the localization density within clusters (*Figure 1H*). These data imply that the distribution and nano-organization of NiV-F are tightly regulated and may be important for membrane fusion activation. We also analyzed NiV-F distribution in different cell lines. HeLa cells express ephrinB2 and -B3, and thus are NiV permissive cells. The overall NiV-F nano-organization seems quite similar on PK13 and HeLa cells, agreeing with our previous results (*Figure 1—figure supplement 2A*; *Liu et al., 2018*). The Hopkins' indices are also similar for NiV-F on PK13 and HeLa cells (*Figure 1—figure supplement 2B*). The DBSCAN analysis shows that the percentage localizations in clusters (*Figure 1—figure supplement 2C*), the relative density defined as the ratio of the localization density in clusters to the total localization density in the selected region (*Figure 1—figure supplement 2D*), and cluster diameters (*Figure 1—figure supplement 2E*) are comparable for NiV-F on PK13 and HeLa cells. These results suggest that NiV-F clusters are minimally affected by the cell lines of expression or the presence of the ephrinB2 and/or -B3.

## The NiV-F nanoclusters are not affected by NiV-F cleavage

NiV-F is synthesized in host cells as inactive precursor $F_0$ and cleaved by cellular proteases cathepsins L and B in the endosomes at an acidic pH to generate the fusion-active, disulfide-linked $F_1$–$F_2$ construct. The $F_1$–$F_2$ is subsequently recycled to the cell surface to induce cell–cell fusion and incorporated into virus particles. We hypothesized that the $F_0$ precursor and the $F_1$–$F_2$ active forms co-existed in the same clusters on the cell surface. To test this hypothesis, we used a pan-cysteine cathepsin inhibitor E64d to inhibit the NiV-F cleavage. E64d inhibited the cell–cell fusion induced by NiV-F and NiV-G in

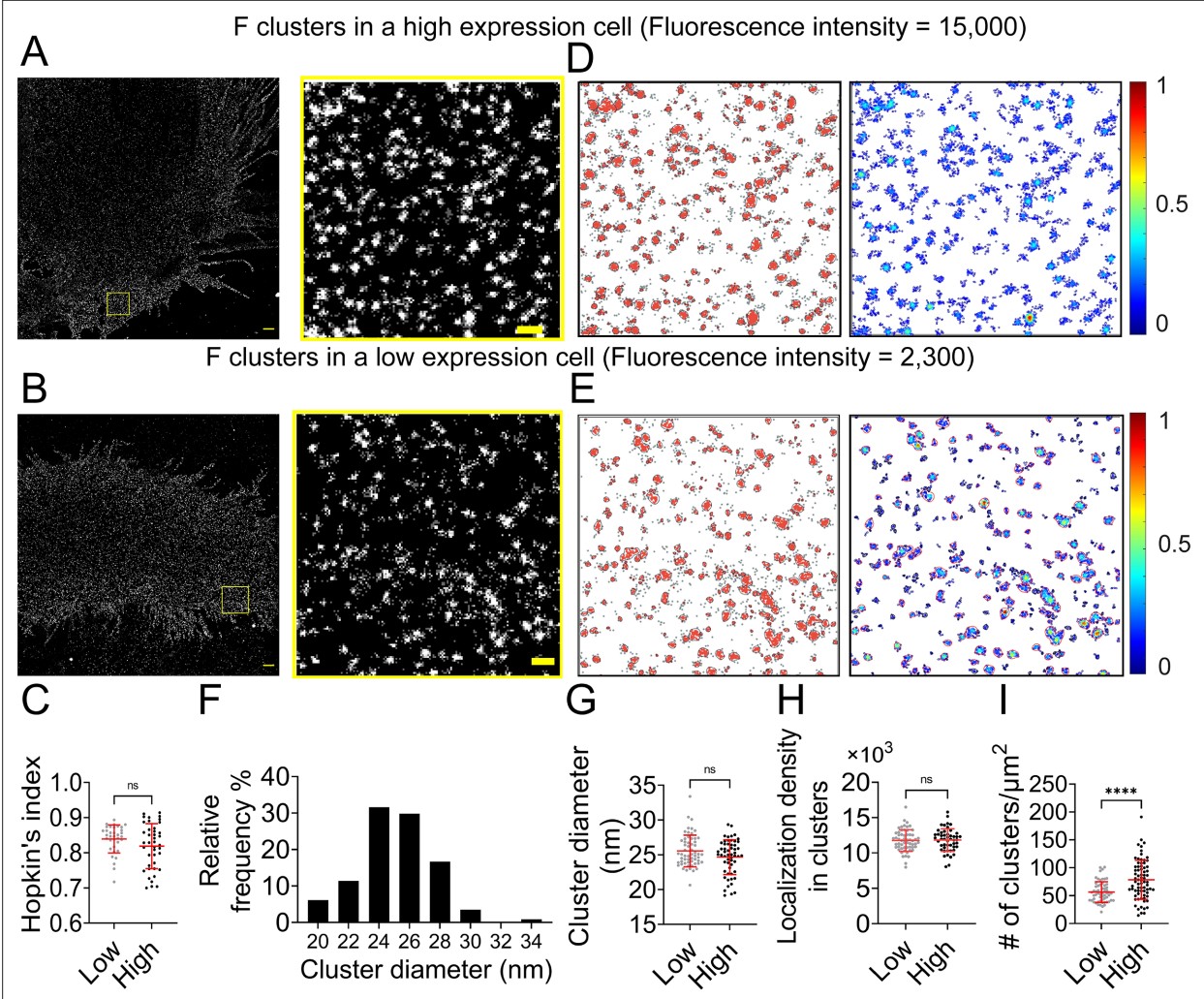

**Figure 1.** NiV-F forms regular-sized clusters that are not affected by the surface expression level. Cross-section ($\Delta z$ = 600 nm) of single-molecule localization microscopy (SMLM) images of NiV-F in high-expression (**A**) and low-expression (**B**) PK13 cells. Scale bar: 1 μm. The yellow boxed region is enlarged to show the detailed distribution pattern. Scale bar: 0.2 μm. (**C**) Hopkin's index of the F localizations in low- and high-expression cells. p = 0.0927; n = 35 and 40. (**D, E**) Cluster maps (left) and localization density maps (right) of the enlarged regions in A and B. Cluster contours are highlighted with gray lines. Normalized relative density is pseudocolored according to the scale on the right. (**F**) The percentage distribution of the NiV-F cluster diameter from 13 cells. n = 114. (**G**) The cluster diameters in low- and high-expression cells. p = 0.2739; n = 58 and 56. (**H**) The localization density (# of localizations per μm²) within clusters in low- and high-expression cells. p = 0.7602; n = 58 and 50. (**I**) The number of clusters per μm² in low- and high-expression cells. p < 0.0001; n = 59 and 74. The cut-off fluorescence intensity for low- and high-expression cells is 8000 (Arb. Unit). Sample size n is the number of total regions from six to eight cells. Bars represent mean ± SD. p value was obtained using the Mann–Whitney test. ns: p > 0.05; ****p < 0.0001.

The online version of this article includes the following source data and figure supplement(s) for figure 1:

**Source data 1.** Related to *Figure 1A*.

**Source data 2.** Related to *Figure 1B*.

**Source data 3.** Related to *Figure 1D*.

**Source data 4.** Related to *Figure 1E*.

**Source data 5.** Related to *Figure 1C–I*.

**Figure supplement 1.** A comparison of clusters formed by NiV-F-FLAG and NiV-F-HA on PK13 cells and resolved by single-molecule localization microscopy (SMLM).

**Figure supplement 1—source data 1.** Related to *Figure 1—figure supplement 1C*.

**Figure supplement 1—source data 2.** Related to *Figure 1—figure supplement 1D*.

**Figure supplement 1—source data 3.** Related to *Figure 1—figure supplement 1E*.

*Figure 1 continued on next page*

*Figure 1 continued*

**Figure supplement 2.** The nanoscale organization of NiV-F is similar in PK13 and HeLa cells.

**Figure supplement 2—source data 1.** Related to *Figure 1—figure supplement 2A*.

**Figure supplement 2—source data 2.** Related to *Figure 1—figure supplement 2B–E*.

HeLa cells (*Figure 2—figure supplement 1*), agreeing with the previous fusion inhibitory effects in Vero and MDCK (Madin-Darby canine kidney) cells (*Diederich et al., 2012*; *Diederich et al., 2005*). Our SMLM images show that E64d treatment does not result in any significant change in the clustering of NiV-F in HeLa cells (*Figure 2A*). The clustering extent of NiV-F was not altered by the E64d treatment, as shown by comparable Hopkin's indices (*Figure 2B*). Additionally, quantitative parameters of the nano-organization such as percentage of localizations in clusters (*Figure 2C*), relative density (*Figure 2D*), cluster size (*Figure 2E*), and the total density of the regions (*Figure 2F*) were comparable. Our data show that clusters formed by non-cleaved NiV-F in cells treated by E64d are similar to those formed by a mixture of cleaved and non-cleaved NiV-F in the control cells, suggesting that F cleavage does not alter F clustering on the cell membrane. These data support our hypothesis that the non-cleaved precursor $F_0$ and cleaved $F_1$–$F_2$ co-exist in the same cluster on the cell membrane.

## Mutations destabilizing the NiV-F hexamer-of-trimer assembly alter its nano-organization on the plasma membrane

We noticed that the uniformity in the NiV-F cluster morphology resembled the hexameric assembly of soluble NiV-F decorated by GCN4 (*Xu et al., 2015*). Next, we investigated whether NiV-F clusters were stabilized by the hexameric interface. As reported, L53D and V108D mutants destabilize the hexameric interface and demonstrate decreased membrane fusion ability; Q393L stabilizes the hexameric interface and demonstrates increased fusion ability (*Xu et al., 2015*). We inserted the FLAG tag into the ectodomains of these mutants at the same position as that of the NiV-F. We verified that the relative fusion ability to their F-WT counterpart was comparable between the FLAG-tagged and untagged mutants (*Figure 3—figure supplement 1A, B*; *Xu et al., 2015*). The FLAG-tagged mutants showed similar CSE levels to NiV-F (*Figure 3—figure supplement 1C*). All mutants showed some levels of processing, although the cleavage of L53D was the least efficient compared to F-WT and other mutants (*Figure 3—figure supplement 1D*). Additionally, the FLAG-tagged NiV-F mutants showed a similar trend in virus entry as that of cell–cell fusion when pseudotyped to a recombinant VSV virus (*Figure 3—figure supplement 1E, F*), agreeing with the previous results obtained using the NiV-F constructs with a C-terminal AU1 tag (*Xu et al., 2015*). We also generated VLPs (virus-like particles) expressing NiV-M-β-lactamase (NiV-M-Bla), NiV-G-HA, and NiV-F-FLAG or mutants for an entry kinetics assay. The entry kinetics was measured in real-time by conversion of green to blue fluorescence resulting from CCF2-AM dye cleavage by the β-lactamase upon the entry of VLPs (*Wolf et al., 2009*; *Landowski et al., 2014*). Our results show that the hyperfusogenic mutant Q393L shows more efficient VLP entry than that of WT, while VLPs bearing the hypofusogenic mutant L53D and V108D are less efficient in entry than that of the WT (*Figure 3—figure supplement 1G, H*). It is noteworthy that the incorporation of NiV-G seems to be impacted by the expression of L53D in both VSV/NiV pseudovirions and VLPs (*Figure 3—figure supplement 1F, H*), and the deficient virus entry induced by L53D may be partly due to the decreased attachment to the target cells (*Figure 3—figure supplement 1E, G*). Nonetheless, the level of L53D on the pseudovirions and VLPs seems comparable to that of the WT (*Figure 3—figure supplement 1F, H*). These data support that mutations at the hexameric interface significantly impact the fusion activity of NiV-F on both cell and virus surface.

NiV-F mutants L53D, V108D, and Q393L were expressed on PK13 cells and subjected to SMLM imaging. The images show that clusters formed by L53D and V108D are smaller and more dispersed than those of the F-WT and Q393L (*Figure 3A*). The Hopkin's index for Q393L is significantly higher than that of F-WT, while the Hopkin's indices for L53D and V108D are slightly lower than that of F-WT (*Figure 3B*). Consistently, a lower percentage of L53D and V108D localizations are segregated into clusters than that of F-WT, while a similar percentage of Q393L and F-WT localizations form clusters (*Figure 3C*). In addition, clusters of L53D and V108D are less packed than that of F-WT, with 48-, 49-, and 55-fold denser than the total localization density in the region, respectively (*Figure 3D*). The clusters of Q393L are not significantly denser than that of F-WT (*Figure 3D*). Our analysis also shows that

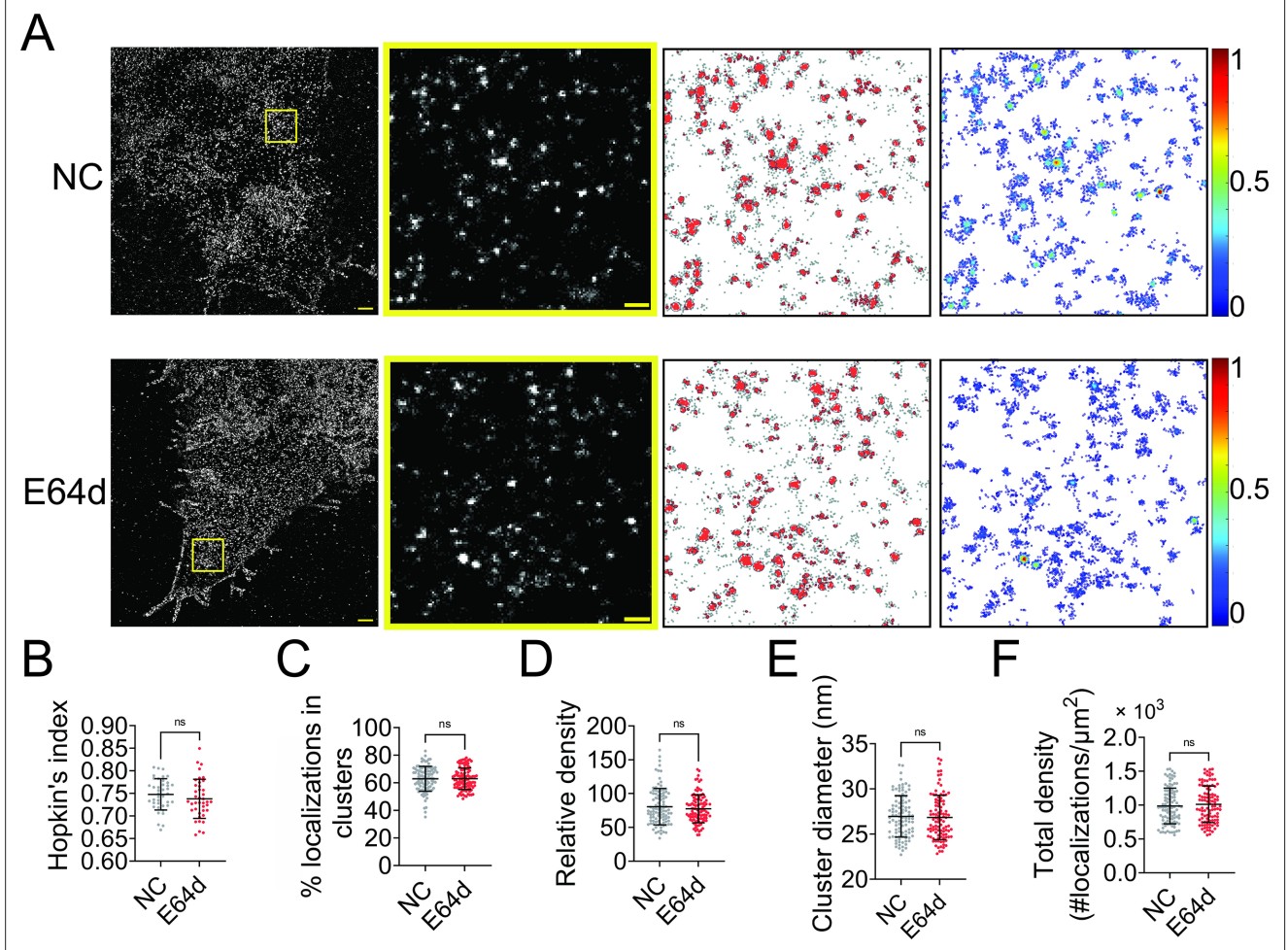

**Figure 2.** Endosomal cleavage does not affect the nanoscale distribution of NiV-F. (**A**) First column: Cross-section (Δz = 600 nm) of single-molecule localization microscopy (SMLM) images of NiV-F in HeLa cells untreated (NC) and treated with 20 μM E64d (E64d). HeLa cells were co-transfected by expression plasmids coding for NiV-G and NiV-F. Twenty μM E64d or the same volume of solvent methanol was added to cells at 2 hr post-transfection. Scale bar: 1 μm. Second column: The yellow boxed region in the first column is enlarged to show individual clusters. Scale bar: 0.2 μm. Third column: Cluster maps from the enlarged regions. Fourth column: Localization density maps show the normalized relative density of the enlarged regions. Quantitative analyses of NiV-F clusters formed in HeLa cells without (NC) and with (E64d) the E64d treatment: (**B**) Hopkin's index, p = 0.1774; n = 40 and 40; (**C**) percentage of localizations in clusters, p = 0.7343; n = 101 and 101; (**D**) relative density, p = 0.6878; n = 100 and 103; (**E**) average cluster diameters, p = 0.5769; n = 100 and 101; (**F**) total density of the region, p = 0.5439; n = 101 and 102. Bars represent mean ± SD. p value was obtained using Mann–Whitney test. ns: p > 0.05. Sample size n is the number of total regions from 4 to 10 cells.

The online version of this article includes the following source data and figure supplement(s) for figure 2:

**Source data 1.** Related to *Figure 2A*.

**Source data 2.** Related to *Figure 2B–F*.

**Figure supplement 1.** E64d treatment inhibits cell–cell fusion induced by NiV-F and -G.

**Figure supplement 1—source data 1.** Related to *Figure 2—figure supplement 1A*.

**Figure supplement 1—source data 2.** Related to *Figure 2—figure supplement 1B*.

L53D and V108D form smaller clusters compared to the F-WT (*Figure 3E*). The Q393L clusters are of a similar size to that of F-WT (*Figure 3E*). The total density of localizations and the total CSE levels for all constructs are comparable (*Figure 3F*, *Figure 3—figure supplement 1C*), suggesting that differences in clustering are not due to variable numbers of localizations or levels of expression. Collectively, these data show that the mutations destabilizing the hexameric interface (L53D and V108D) make NiV-F localizations more dispersed and form smaller and less packed clusters. Q393L that stabilizes the hexameric interface promotes the clustering of NiV-F localizations (*Figure 3B*) but does not

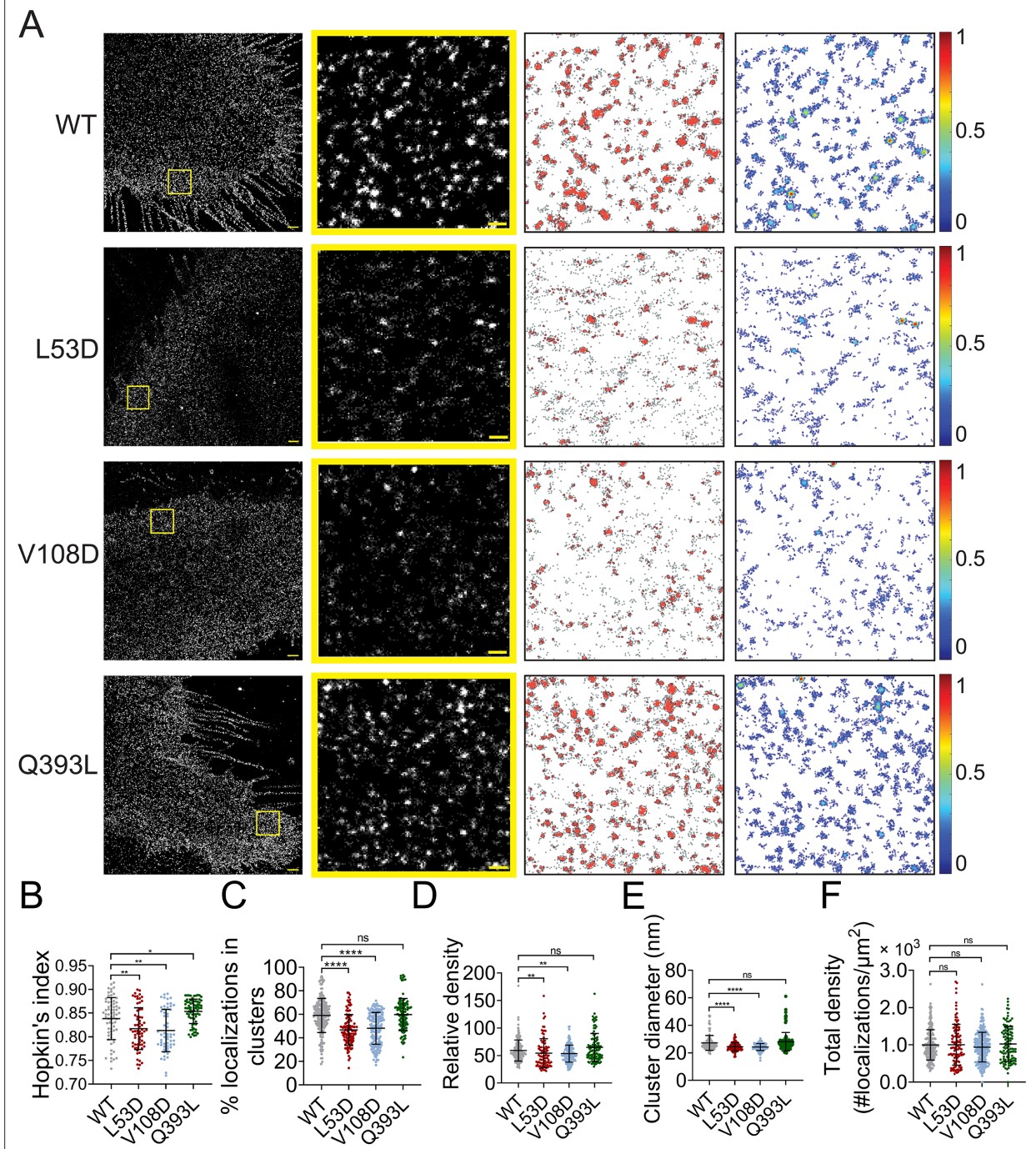

**Figure 3.** Mutations at the NiV-F hexameric interface affect its nano-organization. (**A**) First column: Cross-section (Δz = 600 nm) of single-molecule localization microscopy (SMLM) images of the FLAG-tagged NiV-F-WT (WT), L53D, V108D, and Q393L on PK13 cell membrane. Scale bar: 1 μm. Second column: The yellow boxed regions in the first column are enlarged to show individual clusters. Scale bar: 0.2 μm. Third column: Cluster maps from enlarged regions. Fourth column: Localization density maps show normalized relative density of the enlarged regions. Quantitative analyses of the distribution of the FLAG-tagged NiV-F constructs: (**B**) Hopkin's index, p = 0.0034, 0.0029, and 0.0448; n = 57–70; (**C**) percentage of localizations in clusters, p < 0.0001, <0.0001, and =0.9880; n = 106–198; (**D**) relative density, p = 0.0015, 0.0100, and 0.2244; n = 90–187; (**E**) average cluster diameters, p < 0.0001, <0.0001, and =0.0617; n = 106–198; (**F**) total density of the region (a ratio of total localizations in a region to the size of the region), p = 0.5726, 0.3097, and 0.8209; n = 106–243. Bars represent mean ± SD. Sample size n is the number of total regions from 11 to 20 cells. p value was obtained using Mann–Whitney test. ns: p > 0.05; *p < 0.05; **p < 0.01; ****p < 0.0001.

The online version of this article includes the following source data and figure supplement(s) for figure 3:

**Source data 1.** Related to *Figure 3A*.

*Figure 3 continued on next page*

*Figure 3 continued*

**Source data 2.** Related to *Figure 3B*.

**Source data 3.** Related to *Figure 3C–F*.

**Figure supplement 1.** The fusion ability, expression levels, and processing of FLAG-tagged NiV-F and hexameric mutants.

**Figure supplement 1—source data 1.** Related to *Figure 3—figure supplement 1A*.

**Figure supplement 1—source data 2.** Related to *Figure 3—figure supplement 1B*.

**Figure supplement 1—source data 3.** Related to *Figure 3—figure supplement 1C*.

**Figure supplement 1—source data 4.** Related to *Figure 3—figure supplement 1D*.

**Figure supplement 1—source data 5.** Related to *Figure 2—figure supplement 1D*.

**Figure supplement 1—source data 6.** Related to *Figure 3—figure supplement 1E*.

**Figure supplement 1—source data 7.** Related to *Figure 3—figure supplement 1F*.

**Figure supplement 1—source data 8.** Related to *Figure 3—figure supplement 1F*.

**Figure supplement 1—source data 9.** Related to *Figure 3—figure supplement 1G*.

**Figure supplement 1—source data 10.** Related to *Figure 3—figure supplement 1H*.

**Figure supplement 1—source data 11.** Related to *Figure 3—figure supplement 1H*.

significantly alter the organization and morphology of individual clusters (*Figure 3C–F*). These data suggest that the NiV-F clusters are susceptible to mutations at the hexameric interface. Considering that the uniformed cluster size is similar to the assemblies formed by the soluble GCN4-NiV-F, we propose that NiV-F may organize into hexamer-of-trimer on the plasma membrane.

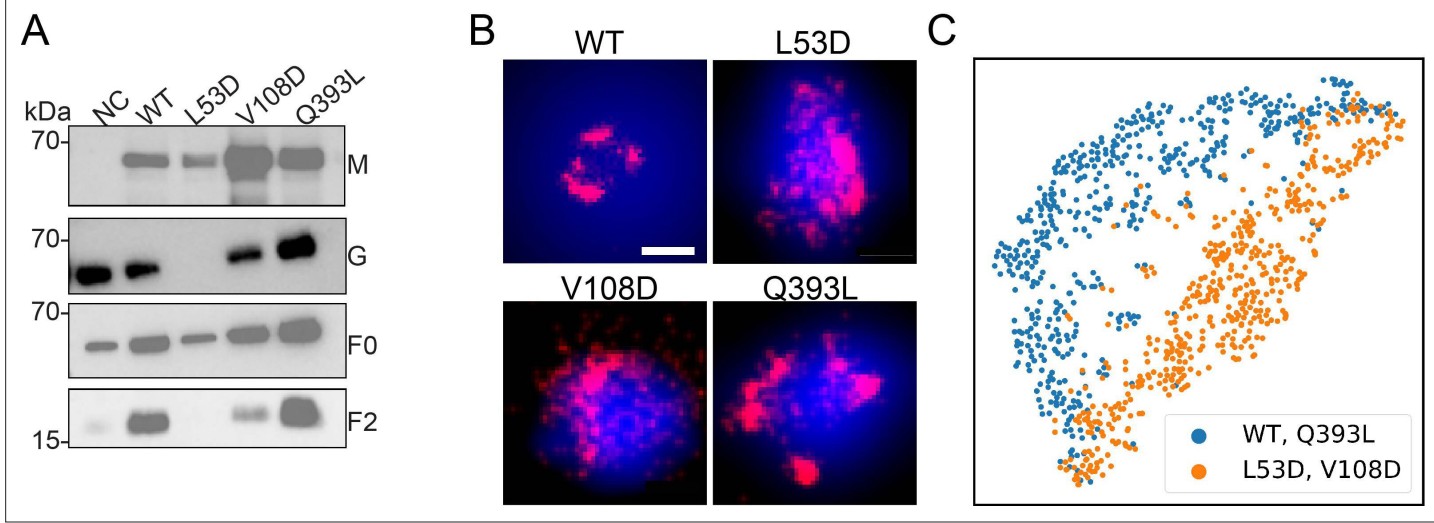

**Figure 4.** The distribution and organization of NiV-F constructs in VLPs. (**A**) The incorporation of F-WT and mutants in VLPs. NiV-M-GFP, G-HA, and FLAG-tagged F-WT or mutants were transfected to 293T cells. The supernatants were collected at 48 hr post-transfection and analyzed on sodium dodecyl sulfate–polyacrylamide gel electrophoresis (SDS–PAGE) followed by western blotting. NiV-M was probed by polyclonal goat anti-GFP, NiV-G polyclonal rabbit anti-HA, $F_0$ and $F_2$ M2 monoclonal mouse anti-FLAG antibody. (**B**) Cross-section ($\Delta z$ = 100 nm) of single-molecule localization microscopy (SMLM) images of the FLAG-tagged NiV-F-WT (WT), L53D, V108D, and Q393L on individual VLPs. Scale bar: 0.2 μm. (**C**) The classification of the ordered sequence of reachability distances of the NiV-F constructs localizations. Orange: F-WT ($n$ = 306) and Q393L ($n$ = 323); blue: L53D ($n$ = 310) and V108D ($n$ = 329). $n$ is the number of VLPs used for classification analysis.

The online version of this article includes the following source data and figure supplement(s) for figure 4:

**Source data 1.** Related to *Figure 4A*.

**Source data 2.** Related to *Figure 4A*.

**Source data 3.** Related to *Figure 4B*.

**Source data 4.** Related to *Figure 4C*.

**Figure supplement 1.** The OPTICS algorithm identifies the clusters of NiV-F-WT and constructs on 3D VLPs.

## Mutations at the NiV-F hexameric interface affect its distribution on the VLP membrane

To further investigate the organization of F mutants in viral membranes, we imaged the FLAG-tagged NiV-F constructs on VLPs produced by 293T cells expressing NiV-M-GFP, NiV-G, and NiV-F-WT, L53D, V108D, or Q393L mutants. The incorporation of NiV-F is comparable among the NiV-F constructs (*Figure 4A*). The L53D and V108D on VLPs are less cleaved compared to that of the F-WT and Q393L, as suggested by the weaker NiV-F$_2$ bands (*Figure 4A*). Notably, the incorporation of NiV-G is largely abrogated in the VLPs that expressing L53D (*Figure 4A*).

To probe the organization of NiV-F constructs on VLPs, the F-constructs on VLP membranes were stained using Alexa Fluor 647 and subjected to SMLM imaging. The GFP fluorescence on NiV-M was used to locate the VLPs. *Figure 4B* shows that F-WT and F-Q393L form distinctive clusters, while localizations of L53D and V108D are more dispersed on a *z* = 100 nm projection. This agrees with their distributions on the plasma membrane (*Figure 3A*). To gain a quantitative insight into the distribution of the F constructs on the three-dimensional (3D) VLPs, we used an algorithm named OPTICS (Ordering Points To Identify Clustering Structure) (*Ankerst et al., 1999*). To identify clusters, the OPTICS algorithm sorts out the localizations by calculating the reachability distance between two adjacent localizations in a propagative manner and generates a sequence of the ordered localizations (*Ankerst et al., 1999*). In the plot of reachability distance vs. the sequence of ordered localizations, a sudden increase in the reachability distance marks the cutoff of a cluster (*Figure 4—figure supplement 1A*). The shade between neighboring peaks represents localizations that are classified in one cluster (*Figure 4—figure supplement 1A*). OPTICS can identify sub-clusters on a 3D VLP that may not be recognized by DBSCAN that uses one fixed parameter for the entire dataset (*Figure 4—figure supplement 1B*) shows the OPTICS plots of the localizations of F-WT and mutants on representative VLPs (*Figure 4B*). The noticeable kinks suggest that F-WT and Q393L form distinctive clusters on VLPs, while the irregular, small kinks indicate that L53D and V108D are more dispersed and form smaller clusters on VLPs (*Figure 4B*, *Figure 4—figure supplement 1B*). To gain a population insight on the nano-organization of the F constructs on VLPs, we developed a one-dimensional convolutional neural network (1D CNN) to classify the ordered sequence of reachability distances of the NiV-F localizations obtained by the OPTICS algorithm (*Kiranyaz et al., 2021*). The distribution patterns of the localizations of the F-WT and Q393L partition into one category, and L53D and V108D in another (*Figure 4C*). These results indicate that L53D and V108D form smaller and more dispersed clusters than F-WT and Q393L on the VLP membranes, agreeing with that of the plasma membrane (*Figure 3*). Our results indicate that the trimer-trimer interaction at the hexameric interface is key in stabilizing the nano-organization of NiV-F on both cell and viral membranes, and NiV-F do not seem to rearrange during the incorporation into VLPs. Nonetheless, our data do not rule out the possibility that the association of nucleocapsids with the plasma membrane during assembly may reorganize NiV-F on the authentic virus membranes. Since the incorporation levels of NiV-M and -G varies among mutants, the nano-organization of NiV-F is likely associated with the virion incorporation or budding.

## The NiV-F clusters are dispersed upon the disruption of the interactions between TMDs

Evidence shows that TMD of class I fusion protein can self-associate in the absence of the rest of the protein and is important for membrane fusion (*Barrett and Dutch, 2020*). Previous studies show that mutations in the Leucine–Isoleucine Zipper (LI zipper) motif lead to the dissociation of an HeV-F TMD-derived peptide, decreased stability, and the fusion ability of the whole HeV-F protein (*Webb et al., 2017*). As the TMD domains of the NiV- and HeV-F demonstrate a 94% similarity, we mutated leucine (488 and 509) and isoleucine (495 and 502) residues of the LI zipper in NiV-F TMD to alanine (LI4A) (*Figure 5—figure supplement 1A*). Similar to their counterparts of HeV-F, the NiV-F-LI4A showed a decreased F processing (*Figure 5—figure supplement 1B*), cell–cell fusion activity (*Figure 5—figure supplement 1C, D*), and CSE levels (*Figure 5—figure supplement 1E*; *Webb et al., 2017*). The NiV-F-LI4A failed to incorporate into VSV/NiV pseudovirions (*Figure 5—figure supplement 1F*), and thus did not induce any virus entry (*Figure 5—figure supplement 1G*). Visual inspection of the images suggests that LI4A forms bigger clusters than F-WT (*Figure 5A*). However, the LI4A seems less likely to cluster than F-WT, indicated by a lower Hopkin's index (*Figure 5B*). Similarly, the portion of LI4A localizations partitioned into clusters is slightly lower than that of WT (*Figure 5C*). Interestingly, we

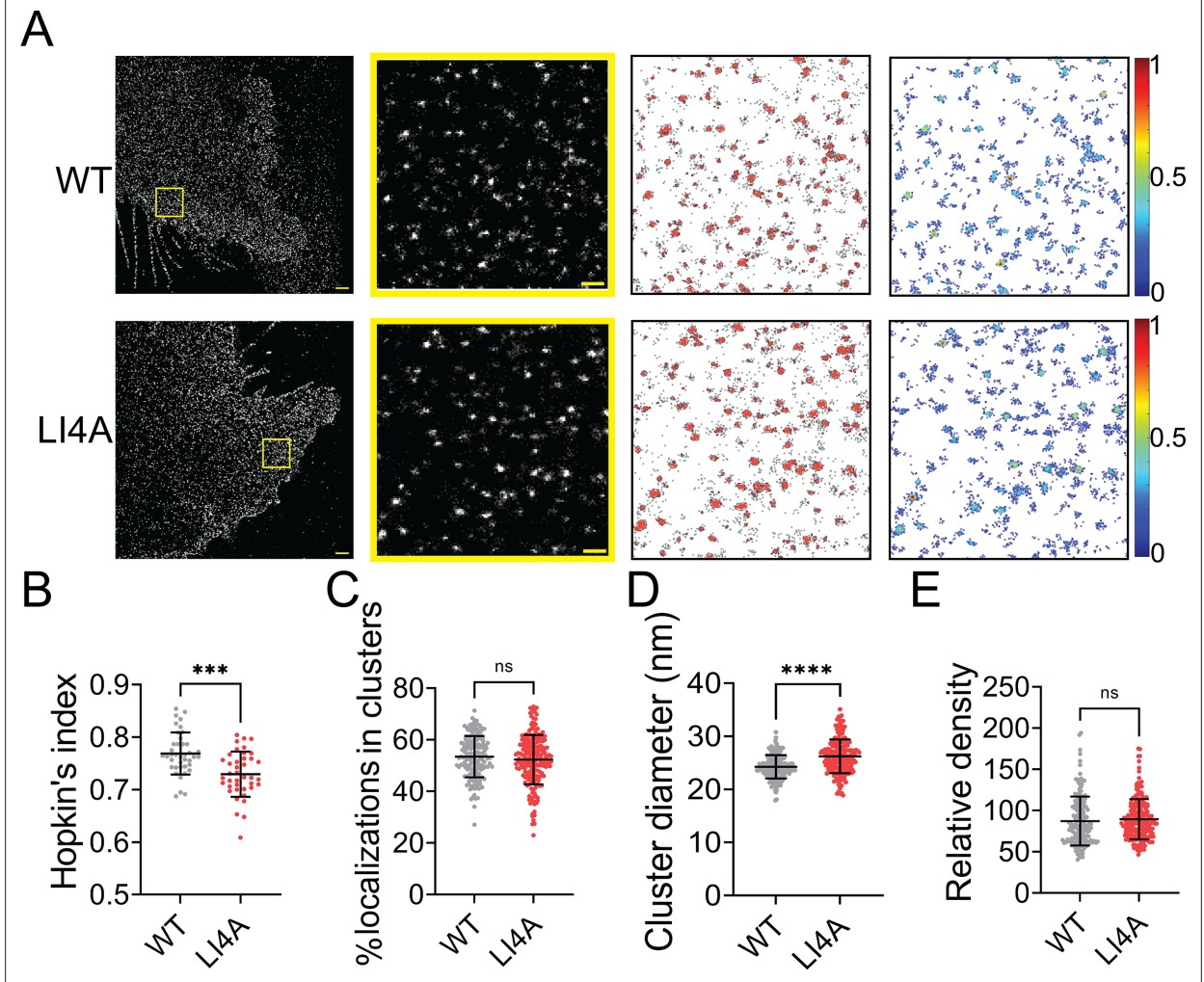

**Figure 5.** Mutations in the LI zipper of the NiV-F transmembrane domain disturb the NiV-F distribution. (**A**) First column: Cross-section (Δz = 600 nm) of single-molecule localization microscopy (SMLM) images of the FLAG-tagged NiV-F-WT (WT) and NiV-F-LI4A (LI4A) mutant on PK13 cell membrane. Scale bar: 1 μm. Second column: The yellow boxed region in the first column is enlarged to show individual clusters. Scale bar: 0.2 μm. Third column: Cluster maps from enlarged regions. Fourth column: Localization density maps show normalized relative density of the enlarged regions in the second column. Quantitative analyses of the WT and LI4A clusters: (**B**) Hopkin's index, p = 0.0001; n = 40 and 40 (**C**) Percentage of localizations in clusters, p = 0.3058; n = 171 and 211; (**D**) Average cluster diameters, p < 0.0001; n = 171 and 210; (**E**) relative density, p = 0.1092; n = 166 and 211. Bars represent mean ± SD. p value was obtained using Mann–Whitney test. ns: p > 0.05; ***p = 0.0001; ****p < 0.0001. Sample size n is the number of total regions from 13 to 16 cells.

The online version of this article includes the following source data and figure supplement(s) for figure 5:

**Source data 1.** Related to *Figure 5A*.

**Source data 2.** Related to *Figure 5B*.

**Source data 3.** Related to *Figure 5C–E*.

**Figure supplement 1.** The NiV-F LI4A does not induce cell–cell fusion.

**Figure supplement 1—source data 1.** Related to *Figure 5—figure supplement 1B*.

**Figure supplement 1—source data 2.** Related to *Figure 5—figure supplement 1B*.

**Figure supplement 1—source data 3.** Related to *Figure 5—figure supplement 1C*.

**Figure supplement 1—source data 4.** Related to *Figure 5—figure supplement 1D, E*.

**Figure supplement 1—source data 5.** Related to *Figure 5—figure supplement 1F*.

**Figure supplement 1—source data 6.** Related to *Figure 5—figure supplement 1F*.

**Figure supplement 1—source data 7.** Related to *Figure 5—figure supplement 1G*.

noticed that clusters formed by LI4A are significantly bigger (*Figure 5D*) than that of F-WT and with a similar localization density within the clusters (*Figure 5E*). The data suggest that interactions between TMDs of NiV-F monomers play a role in stabilizing its nano-organization. Dissociation of TMDs by mutations may result in increased space between F monomers and thus leads to bigger clusters on the cell membrane.

## The NiV-F nanoclusters are stabilized by endocytosis components

NiV-F can be endocytosed for endosomal cleavage. Next, we investigated whether NiV-F clusters are stabilized during endocytosis. NiV-F contains an endocytosis sorting signal YSRL and an additional YY motif at its cytoplasmic tail (*Figure 6—figure supplement 1A*). Mutation of these tyrosine residues to alanine almost diminished NiV-F cleavage (*Diederich et al., 2005*). An FLAG-tagged NiV-F-YA construct containing the aforementioned mutations resulted in significantly less $F_2$ band than F-WT (*Figure 6—figure supplement 1B*) and completely abrogated cell–cell fusion in 293T cells (*Figure 6—figure supplement 1C, D*), although the CSE levels were comparable to that of F-WT (*Figure 6—figure supplement 1E*). The YA mutant did not incorporate into the VSV/NiV pseudovirions (*Figure 6—figure supplement 1F*) and thus did not mediate virus entry (*Figure 6—figure supplement 1G*). By visually inspecting the SMLM images, we noticed that F-YA was less clustered than that of F-WT (*Figure 6A*). Indeed, the Hopkin's analysis suggests that F-YA is less clustered than F-WT (*Figure 6B*). The DBSCAN analysis shows that lower percentage of F-YA localizations segregate into clusters (*Figure 6C*) than that of the F-WT, and clusters formed by F-YA are smaller (*Figure 6D*) and less dense (*Figure 6E*) than that of the F-WT. Notably, there is no significant difference in the total density for F-YA and F-WT SMLM images (*Figure 6F*), indicating that the differences in cluster organizations do not result from overall protein expression or the stochastic blinking properties of the fluorophore. These results suggest that the endocytosis sorting signal of NiV-F may facilitate the enrichment of NiV-F into clusters on the cell surface, preparing them for endosomal cleavage and/or incorporation into viruses.

It is established that the Yxx$\Phi$ sorting signal on the endocytosis cargo is bound and enriched by the heterotetrameric adaptor protein (AP) complexes, which further recruit the assembly of the clathrin coat (*Bonifacino and Neefjes, 2017*; *Bonifacino and Traub, 2003*). We hypothesize that the clustering of NiV-F is promoted by the assembly of the clathrin coat via the NiV-F–AP-2 interactions. As suggested by a yeast two-hybrid analysis, NiV-F interacts with μ1, μ2, μ3, and μ4 subunits of the AP complexes AP-1, AP-2, AP-3, and AP-4, and mutations of Y525, Y542, and Y543 almost completely abolished the interaction between AP-2 and NiV-F (*Mattera et al., 2014*). Indeed, the co-immunoprecipitation assay shows that less AP2μ2 is pulled down by F-YA than that of F-WT, suggesting that F-YA has a reduced interaction with AP-2 (*Figure 6—figure supplement 1H*). Next, we examined the NiV-F clusters in HeLa cells treated by the endocytosis inhibitor pitstop2. Pitstop2 blocks clathrin-mediated endocytosis by obstructing the binding of the accessory protein (e.g. AP-2) and the clathrin terminal domain, and potentially preventing the assembly of the clathrin coat (*von Kleist et al., 2011*). The SMLM images (*Figure 6G*) show that the NiV-F localizations are more dispersed upon pitstop2 treatment, agreeing with the lower Hopkin's index (*Figure 6H*) and a smaller percentage of localizations in clusters (*Figure 6I*) in pitstop2 treated group compared to that of the control group. Interestingly, we noticed that the NiV-F clusters in pitstop2 treated cells were larger than the control cells (*Figure 6J*), although with similar densities (*Figure 6K*). The total density is consistent between the pitstop2 treated and control groups (*Figure 6L*). The overall dispersed NiV-F distribution upon pitstop2 treatment may potentially be caused by the disrupted assembly of clathrin coat. In combination, these results suggest that the enrichment of NiV-F by AP-2 and the subsequent assembly of the clathrin-coated endosomes are important in stabilizing the F clusters on cell membranes. We envision that the endocytosis components may also facilitate the NiV-F clustering in virus particles as previous proteomic studies show the presence of endocytosis related proteins, such as clathrin, AP-2, and dynamin in NiV VLPs (*Johnston et al., 2019*; *Pentecost et al., 2015*; *Vera-Velasco et al., 2018*).

## Discussion

Here, we used SMLM to resolve the arrangement and organization of NiV-F on the biological membranes at a precision of 10 nm. Our data support the following conclusions: (1) NiV-F is organized

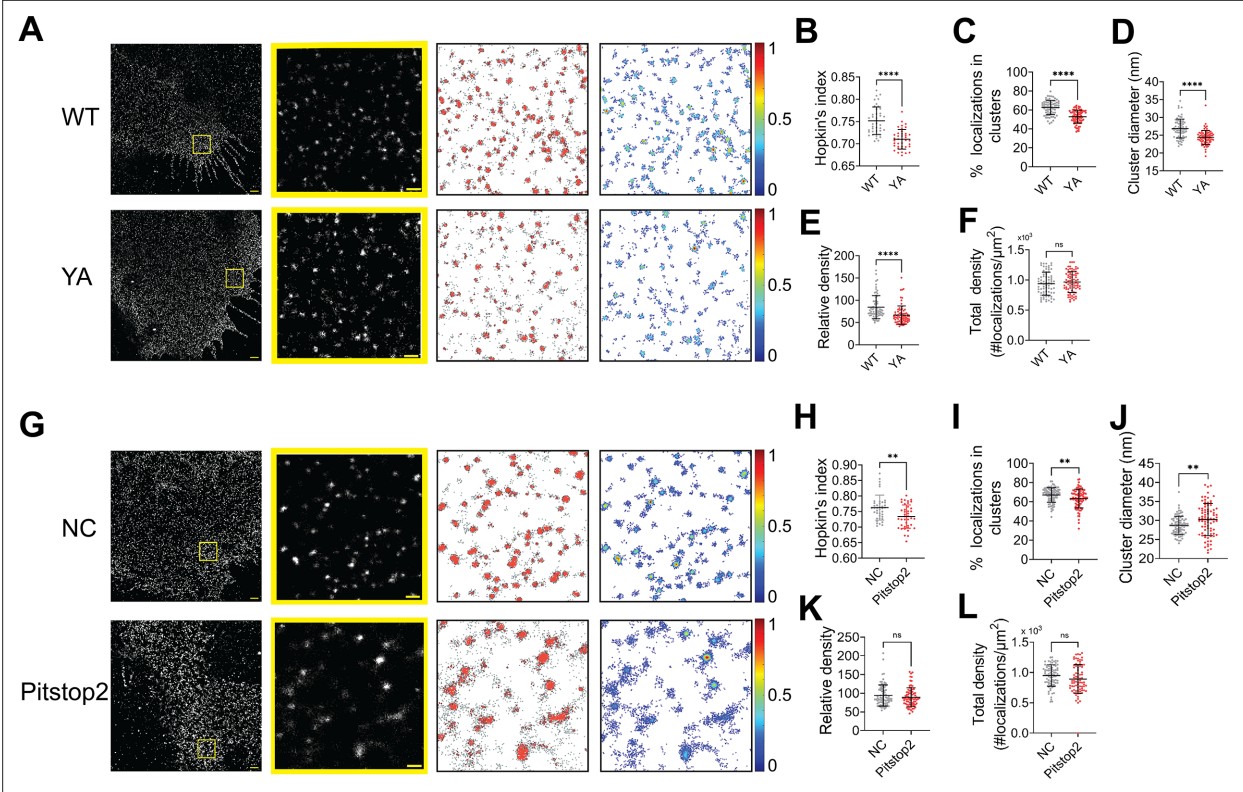

**Figure 6.** The NiV-F nanoclusters are stabilized by endocytosis components. (**A**) First column: Cross-section (Δz = 600 nm) of single-molecule localization microscopy (SMLM) images of the FLAG-tagged NiV-F-WT (WT) and NiV-F-YA (YA) mutant on PK13 cell membrane. Scale bar: 1 µm. Second column: The yellow boxed region in the first column is enlarged to show individual clusters. Scale bar: 0.2 µm. Third column: Cluster maps from enlarged regions. Fourth column: Localization density maps show normalized relative density of the enlarged regions in the second column. Quantitative analyses of the WT and YA clusters: (**B**) Hopkins index of WT and YA, p < 0.0001; n = 40 and 40; (**C**) percentage of localizations in clusters, p < 0.0001; n = 72 and 77; (**D**) average cluster diameters, p < 0.0001; n = 72 and 77; (**E**) relative density, <0.0001; n = 71 and 76; (**F**) total density of the region, p = 0.3560; n = 72 and 77. (**G**) First column: Cross-section (Δz = 600 nm) of SMLM images of the FLAG-tagged NiV-F-WT treated without (NC) and with pitstop2 (pitstop2) on HeLa cell membrane. Scale bar: 1 µm. Second column: The yellow boxed region in the first column is enlarged to show individual clusters. Scale bar: 0.2 µm. Third column: Cluster maps from enlarged regions. Fourth column: Localization density maps show normalized relative density of the enlarged regions in the second column. Quantitative analyses of NiV-F without (NC) and with pitstop2 (pitstop2): (**H**) Hopkins index, p = 0.0054; n = 40 and 40; (**I**) percentage of localizations in clusters, p = 0.0060; n = 82 and 85; (**J**) average cluster diameters, p = 0.0057; n = 82 and 81; (**K**) relative density, p = 0.4235; n = 79 and 78; (**L**) total density of the region, p = 0.0607; n = 82 and 85. Bars represent mean ± SD. p value was obtained using Mann–Whitney test. ns: p > 0.05; **p < 0.01; ****p < 0.0001. Sample size n is the number of total regions from four to nine cells.

The online version of this article includes the following source data and figure supplement(s) for figure 6:

**Source data 1.** Related to *Figure 6A*.

**Source data 2.** Related to *Figure 6B*.

**Source data 3.** Related to *Figure 6C–F*.

**Source data 4.** Related to *Figure 6G*.

**Source data 5.** Related to *Figure 6H*.

**Source data 6.** Related to *Figure 6I–L*.

**Figure supplement 1.** The FLAG-tagged NiV-F-YA mutant inhibits NiV-F cleavage, cell–cell fusion, and the NiV-F-AP-2 interaction in 293T cells.

**Figure supplement 1—source data 1.** Related to *Figure 6—figure supplement 1B*.

**Figure supplement 1—source data 2.** Related to *Figure 6—figure supplement 1B*.

**Figure supplement 1—source data 3.** Related to *Figure 6—figure supplement 1C*.

**Figure supplement 1—source data 4.** Related to *Figure 6—figure supplement 1D, E*.

**Figure supplement 1—source data 5.** Related to *Figure 6—figure supplement 1F*.

**Figure supplement 1—source data 6.** Related to *Figure 6—figure supplement 1F*.

*Figure 6 continued on next page*

*Figure 6 continued*

**Figure supplement 1—source data 7.** Related to *Figure 6—figure supplement 1G*.

**Figure supplement 1—source data 8.** Original files for the immunoprecipitation assay.

**Figure supplement 1—source data 9.** PPTX files indicating the relevant bands in *Figure 6—figure supplement 1H*.

into nanoclusters on the biological membranes and this organization is independent of the protein expression level or endosomal cleavage; (2) the NiV-F nano-organization is susceptible to mutations at the trimer interface on the NiV-F ectodomain and the putative oligomerization motif on the TMD; (3) NiV-F sequestered in nanoclusters favors membrane fusion activation; (4) the interactions among NiV-F, the AP-2 complex, and the clathrin coat assembly stabilize the NiV-F clusters. We propose that the NiV-F nanoclusters are the fundamental unit of the NiV fusion machinery, and this organization facilitates membrane fusion triggering by a mixed population of NiV-F molecules with varied degrees of cleavage and opportunities of interacting with NiV-G/receptor complex.

The nano-organization of NiV-F informs the coordinated membrane fusion triggering by a mixed population of NiV-F. This notion is supported by the following observations from this and previous studies: (1) NiV-F molecules form distinctive, regular-sized nanoclusters on the cell and virus membranes (*Figure 1A–E*); (2) the cleaved, active F co-exist with the non-cleaved, inactive F in a nanocluster, as suggested by that the NiV-F nanoclusters are resistant to the cathepsin inhibitor, E64d (*Figure 2*); (3) the NiV-F and G are segregated into different clusters (*Liu et al., 2018*); and (4) the full-length NiV-F and -G proteins do not show stable interactions either before or after ephrinB2 activation (*Wong et al., 2021*). These observations indicate that the NiV-F triggering may be a result of transient interactions between the F molecules at the edge of the clusters with the adjacent ephrinB2/G complex. Moreover, we observed that the NiV-F sequestering into clusters is favorable for membrane fusion activation (*Figures 3–5*), highlighting the importance of the spatial arrangement of NiV-F in its fusion activity. Therefore, it is likely that the triggered, fusion-active NiV-F at the cluster peripheral can facilitate other F molecules in the same cluster to fulfill the conformational changes, and thus maximize the energy to merge the opposing membranes. In contrast, the correlation between the clustering pattern of NiV-F and its incorporation into virions is not obvious. Although the YA and LI4A mutants that showed aberrant clusters cannot incorporate into VSV/NiV pseudovirions (*Figure 5—figure supplement 1F* and *Figure 6—figure supplement 1F*), the clustering-deficient hexameric interface mutants L53D and V108D were incorporated into VLPs (*Figure 4A*) and VSV/NiV pseudovirisons (*Figure 3—figure supplement 1F*), comparable to that of NiV-F-WT. Therefore, the significance of NiV-F clustering for assembly and budding of the virions may be context dependent.

A previous study revealed a hexamer-of-trimer assembly of soluble, GCN4-decorated, prefusion NiV-F. Using SMLM, we estimated that the NiV-F clusters on the cell plasma membrane had a diameter between 24 and 26 nm (*Figure 1F*). This estimation agrees with the size of the hexameric soluble NiV-F-GCN4 assemblies (*Xu et al., 2015*). Notably, the density of clusters (*Figure 1I*), but not the size and localization density of individual clusters (*Figure 1G, H*), is affected by the surface expression levels of NiV-F, implying that the NiV-F clusters are highly regulated. A collaborative effort of multiple copies of viral envelope proteins is favorable for receptor binding and overcoming the energy barrier for membrane fusion. Nanoclusters formed by viral glycoproteins have been observed for HIV-1 (*Chojnacki and Eggeling, 2018*; *Muranyi et al., 2013*), Herpes virus (*Laine et al., 2018*), and influenza virus (*McMahon et al., 2023*). The viral restriction factor, serine incorporator protein 5 (SERINC5), inhibits HIV-1 fusion by disrupting the HIV env nanoclusters, highlighting the importance of the nanoclusters in the function of the viral glycoproteins (*Chen et al., 2020*). More interestingly, the NiV-F distribution becomes more dispersed, characterized by smaller and less dense clusters, upon the mutation of two key residues, L53 and V108, at the hexameric interface, suggesting that the NiV-F clusters are likely to be the hexamer-of-trimer assemblies on the cell and viral membranes (*Figures 3 and 4*).

The hexamer-of-trimer NiV-F is observed on the VLP surface by electron tomography (*Xu et al., 2015*). The NiV-F hexamer-of-trimers are arranged into a soccer ball-like structure, with one trimer being part of multiple hexamer-of-trimers. Nonetheless, the hexagonal lattice has been reported for heamagglutinin-esterase-fusion (HEF) proteins on the surface of the influenza C virus (*Halldorsson et al., 2021*) and Env glycoprotein on the Foamy virus (*Effantin et al., 2016*). The sharing of one

trimer among hexamer-of-trimers and conformational arrangements in the ectodomain of viral glycoproteins were shown to stabilize the contacts for lattice formation (*Xu et al., 2015*; *Halldorsson et al., 2021*; *Effantin et al., 2016*). In our study, the NiV-F clusters observed on the cell plasma membrane are more likely to be individual hexamer-of-trimers estimated by size (*Figure 1*). We speculate that the formation of the hexagonal lattice may be a result of increased concentration of NiV-F on VLP membrane. A study on measles virus shows only limited high order of MeV-F assemblies at some virus assembly sites in recMeV-(H-118 ∇41×) cells (*Ke et al., 2018*), however, we have not observed increased NiV-F localizations at NiV-M-positive membrane domains on the cell body of NiV-M, F, and G co-expression PK13 cells (*Liu et al., 2015*). Therefore, the formation of NiV-F lattice on the VLP membrane may occur either at a late stage of virus assembly or after budding. Additionally, cellular factors may also play a role in maintaining individual NiV-F hexamer-of-trimers on the plasma membrane, as a proteomic analysis reveals that NiV-F is an interacting partner of many vesicular trafficking and actin cytoskeleton factors (*Johnston et al., 2019*; *Vera-Velasco et al., 2018*).

We also identified that the endosomal components as key host factors in maintaining the NiV-F nano-organization on the cell membrane. Our data suggest that both the disruption of the F/AP-2 interaction and the inhibition of clathrin assembly by pitstop2 result in dispersed NiV-F distribution. AP-2 is the key adaptor of clathrin-mediated endocytosis. It binds cargo and PtdIns(4,5)P$_2$-containing membrane via multiple interface, and undergoes conformational changes to append to the clathrin lattice (*von Kleist et al., 2011*; *Kovtun et al., 2020*). Our data indicate that AP-2-NiV-F interaction can enrich and strengthen the F nanoclusters (*Figure 6A–F*, *Figure 6—figure supplement 1H*), and potentially facilitate the uptake of NiV-F by endocytosis for endosomal cleavage and activation. It is reported that the clustering of CD44, modulated by *N*-glycosylations, facilitates the uptake of CD44 by endocytosis. Furthermore, the assembly of the clathrin coat at the plasma membrane may also strengthen the F clusters, because pitstop2 that inhibits the clathrin-coat assembly by blocking the clathrin terminal domain (*von Kleist et al., 2011*) prevents the cluster formation of NiV-F. It is plausible that both clathrin and AP-2 facilitate NiV-F clustering on cell and virus membranes because of the similar organization of F on virus and cell membranes and the presence of clathrin and AP-2 in VLPs (*Johnston et al., 2019*; *Vera-Velasco et al., 2018*).

In conclusion, our observations provide direct evidence on the nano-organization and distribution of NiV-F on the cell and viral membranes and shed lights to the fusion activation mechanisms. Therefore, it would be critical to elucidate host factors that maintain NiV-F clusters and the interacting motif in NiV-F, which may represent a new therapeutic strategy for NiV infections. It would also be interesting to analyze the reorganization of the NiV-F clusters upon triggering by the NiV-G/ receptor complex on live cell membranes by combining single-molecule imaging and the supported lipid bilayer technologies (*Wong et al., 2021*). Additionally, a precise stoichiometry of individual NiV-F clusters on the cell and virus membranes could facilitate the vaccine design.

# Materials and methods

## Key resources table

| Reagent type (species) or resource | Designation | Source or reference | Identifiers | Additional information |
|---|---|---|---|---|
| Cell line (*Homo sapiens*) | HEK293T | ATCC | CRL-3216 | |
| Cell line (*Sus scrofa*, pig) | PK13 | ATCC | CRL-6489 | |
| Cell line (*Homo sapiens*) | HeLa | ATCC | CCL-2 | |
| Cell line (*Cercopithecus aethiops*) | Vero | ATCC | CCL-81 | |
| Recombinant DNA reagent | pcDNA 3 plasmids containing cDNA for NiV-F-FLAG and mutants | This paper | | The FLAG tag was inserted after residue 104 of the codon-optimized NiV-F (GenBank accession no. AY816748.1). The hexameric interface mutants (*Figure 3*; *Xu et al., 2015*), LI4A mutant (*Figure 5*; *Webb et al., 2017*), and YA mutant (*Figure 6*; *Diederich et al., 2005*) were constructed as previously published. |

*Continued on next page*

*Continued*

| Reagent type (species) or resource | Designation | Source or reference | Identifiers | Additional information |
|---|---|---|---|---|
| Recombinant DNA reagent | pcDNA 3 plasmid containing cDNA for NiV-F-HA | This paper | | The HA tag was inserted after residue 104 of the codon-optimized NiV-F (GenBank accession no. AY816748.1). |
| Recombinant DNA reagent | pcDNA 3 plasmid containing cDNA for NiV-G-HA | *Liu et al., 2018* | | |
| Recombinant DNA reagent | pcDNA 3 plasmid containing cDNA for NiV-M-GFP | *Liu et al., 2018*; *Wang et al., 2010* | | The GFP gene was fused to the N-terminus of NiV-M gene (GenBank accession no. EU480491.1) |
| Recombinant DNA reagent | pcDNA 3 plasmid containing cDNA for AP2μ2-mCherry | Christien Merrifield | RRID:Addgene_27672 | |
| Recombinant DNA reagent | pcDNA 3 plasmid containing cDNA for untagged NiV-F | *Liu et al., 2018* | | Codon-optimized NiV-F (GenBank accession no. AY816748.1). |
| Recombinant DNA reagent | pcDNA 3 plasmid containing cDNA for NiV-F-AU1 | | | The AU1 tag was inserted at the C terminal of the codon-optimized NiV-F (GenBank accession no. AY816748.1). |
| Recombinant DNA reagent | pCAGGs plasmid containing cDNA for NiV-M-β-lactamase | *Wolf et al., 2009*, *Landowski et al., 2014* | | The β-lactamase gene was fused to the N-terminus of NiV-M gene (GenBank accession no. EU480491.1) |
| Sequence-based reagent | NiV-F-L53D-F1 | This paper | PCR primers | GAGCAACCCCgacACCAAGGACATCGTG |
| Sequence-based reagent | NiV-F-L53D-R1 | This paper | PCR primers | TTGGTgtcGGGGTTGCTCTTGATC |
| Sequence-based reagent | NiV-F-V108D-F2 | This paper | PCR primers | ATAAGGTGGGCGACGacCGGCTGGCCG |
| Sequence-based reagent | NiV-F-V108D-R2 | This paper | PCR primers | gtCGTCGCCCACCTTATCGTCGTCA |
| Sequence-based reagent | NiV-F-Q393L-F3 | This paper | PCR primers | CGTGACCTGCCtGTGCCAGACCAC |
| Sequence-based reagent | NiV-F-Q393L-R3 | This paper | PCR primers | GTGGTCTGGCAcagGCAGGTCACG |
| Sequence-based reagent | NiV-F-FLAG-G/L-F | This paper | PCR primers | GCCTGTGCATCctgCTGATCACCTTC |
| Sequence-based reagent | NiV-F-FLAG-G/L-R | This paper | PCR primers | CAGcagGATGCACAGGCTGGCGATGC |
| Sequence-based reagent | NiV-F-LI-A-F1 | This paper | PCR primers | ATCCTGTACGTGCTGAGCgccGCCAG CCTGTGCATCGGCGccATCACCTTCA |
| Sequence-based reagent | NiV-F-LI-A-R1 | This paper | PCR primers | GCTCAGCACGTACAGGATGgcCATGG ACAGCATGCTGATggcGCTGGGGTTC ACGGTGTC |
| Sequence-based reagent | NiV-F-YA-F | This paper | PCR primers | GCACCCGCCGGTCCTCCAGCCGGCTG gcGGTGTTCCGC |
| Sequence-based reagent | NiV-F-YA-R | This paper | PCR primers | AGGACCGGCGGGTGCGGCCCACCAGC AGCGGCGACCTGGCCGCCATCGGCACCTGATAA |
| Chemical compound, drug | E64d | Sigma-Aldrich | E8640-250UG | 20 μM final concentration for treatment |
| Chemical compound, drug | Pitstop2 | Sigma-Aldrich | SML1169-5MG | 30 μM final concentration for treatment |
| Chemical compound, drug | Tris | EMD Millipore | 648311-1KG | 50 mM for TN buffer |
| Chemical compound, drug | NaCl | Sigma-Aldrich | S9888-1KG | 10 mM for TN buffer |
| Chemical compound, drug | Glucose oxidase | Sigma-Aldrich | G2133-50KU | 0.5 mg/ml for imaging buffer |
| Chemical compound, drug | Catalase | Sigma-Aldrich | C100 | 40 μg/ml for imaging buffer |
| Chemical compound, drug | Mercapto ethylamine (MEA) | Sigma-Aldrich | 30070-10G | 50 mM for SMLM imaging buffer |

*Continued on next page*

*Continued*

| Reagent type (species) or resource | Designation | Source or reference | Identifiers | Additional information |
|---|---|---|---|---|
| Chemical compound, drug | HEPES | Fisher Scientific | 15-630-080 | 25 mM for fusion buffer |
| Chemical compound, drug | Glutamine | Fisher Scientific | 25-030-081 | 2 mM for fusion buffer |
| Chemical compound, drug | $CaCl_2$ | Sigma-Aldrich | C7902 | 1 mM for fusion buffer |
| Chemical compound, drug | Probenecid | Sigma-Aldrich | P8761-25G | 2.5 mM for fusion buffer |
| Chemical compound, drug | Solution D | Fisher Scientific | K1156 | 1:20 for Entry kinetics loading solution |
| Antibody | Anti-FLAG mouse monoclonal | Sigma-Aldrich | F1804 | IF, SMLM: 1:100, Flow: 1:200; WB: 1:500–1:5000 |
| Antibody | Anti-HA rabbit polyclonal antibody | Biolegend | 902301 | IF: 1:900 WB: 1:2000 |
| Antibody | Anti-GFP goat polyclonal antibody | Abcam | Ab5450 | WB: 1:1000 |
| Antibody | Anti-mCherry rabbit polyclonal antibody | Abcam | Ab167453 | WB: 1:2000 |
| Antibody | Anti-β-lactamase mouse monoclonal | Santa Cruz Biotechnology | Sc-66062 | WB: 1:1000 |
| Antibody | Anti-mouse donkey polyclonal antibody, Alexa Fluor 647 conjugated | Invitrogen | A31571 | IF and SMLM: 1:400 |
| Antibody | Anti-rabbit donkey polyclonal antibody, Alexa Fluor 488 conjugated | Invitrogen | A21206 | IF: 1:400 |
| Antibody | Anti-mouse donkey polyclonal antibody, Alexa Fluor 488 conjugated | Invitrogen | A21202 | Flow: 1:400 |
| Antibody | Anti-goat donkey polyclonal antibody, HRP conjugated | Jackson Immunoresearch | 705-035-147 | WB: 1:5000 |
| Antibody | Anti-mouse goat polyclonal antibody, HRP conjugated | Bio-Rad | 1705047 | WB: 1:5000 |
| Antibody | Anti-rabbit goat polyclonal antibody, HRP conjugated | Bio-Rad | 1706515 | WB: 1:5000 |
| Commercial assay or kit | μMACS anti-DYKDDDDK starting kit | Miltenyi Biotec | 130-101-636 | |
| Commercial assay or kit | Renilla Luciferase Assay System | Promega | E2820 | |
| Commercial assay or kit | LiveBLAzer FRET-B/G Loading Kit with CCF2-AM | Invitrogen | K1032 | |
| Commercial assay or kit | QIAamp Viral RNA Kits for RNA Extraction | QIAGEN | 52904 | |
| Commercial assay or kit | SuperScript III First-Strand Synthesis System | Invitrogen | 18080051 | |
| Software | SMLM image reconstruction | *Liu et al., 2018* | | MATLAB Codes are available upon request. |
| Software | ClusDoc | *Pageon et al., 2016* | | Codes were modified to process the data generated in the custom-built SMLM microscope. |
| Software | OPTICS | This paper; *Ankerst et al., 1999* | | C++ Codes are available on GitHub, copy archived at *QLlab, 2024* |
| Software | One-dimensional convolutional neural network (1D CNN) | This paper | | PYTHON Codes are available on GitHub, copy archived at *QLlab, 2024* |
| Software | Prism GraphPad | Dotmatics | RRID:SCR_002798 | |
| Software | Flowjo | BD | RRID:SCR_008520 | |

## Cell lines and plasmids

PK13, HeLa, Vero, and HEK293T cells were cultured at 37°C and 5% $CO_2$ in DMEM (Sigma-Aldrich, D6429) complemented with 10% fetal bovine serum (Invitrogen, 12483-020). Cells were passaged using phosphate-buffered saline (PBS, Invitrogen, 10010-049) and 0.25% Trypsin–EDTA solution (Invitrogen, 25002-072). Cells were monitored routinely for mycoplasma contamination using a mycoplasma detection PCR kit (ABM, G238). An FLAG (NiV-F-FLAG) or HA (NiV-F-HA) tag was inserted after residue 104 of codon-optimized NiV-F in pcDNA3.1 vector. An AU1(NiV-F-AU1) tag was inserted at the C-terminal of codon-optimized NiV-F in pCAGGS vector. The untagged NiV-F does not contain any epitope tag. NiV-F-YA, L53D, V108D, Q393L, and LI4A were produced by site-directed mutagenesis using the NiV-F-FLAG in pcDNA3.1 vector as a template (*Diederich et al., 2005*; *Xu et al., 2015*; *Webb et al., 2017*). AP2μ2-mcherry plasmid is a gift from Christien Merrifield (RRID:Addgene_27672). The HA-tagged NiV-G, GFP-tagged NiV-M, and β-lactamase-NiV-M were constructed previously (*Negrete et al., 2005*; *Wolf et al., 2009*; *Wang et al., 2010*).

## Immunofluorescence for SMLM

For SMLM imaging on cells, $1 \times 10^5$ PK13 or HeLa cells were seeded on coverslips (Marienfeld #1.5 H, 18 mm) coated with 2.5 μg fibronectin (Sigma-Aldrich, F4759-2 mg) in a 12-well plate, and transfected with 1 μg NiV-F variants using lipofectamine 3000 (Invitrogen, L3000015) on the following day. E64d was dissolved in methanol and pitstop2 in DMSO (Dimethylsulfoxide) as recommended by the manufacturer. E64d was added to cells at 3 hr post-transfection for a total treatment time of 15 hr. Pitstop2 was added to cells at 10 hr post-transfection for a total treatment time of 8 hr. At 18 hr post-transfection, cells were fixed with phosphate-buffered saline (PBS) containing 4% paraformaldehyde (PFA; Electron Microscopy Sciences; 50980487) and 0.2% glutaraldehyde (Sigma-Aldrich, G5882-50 ml) for 90 min at room temperature. Cells were treated with signal enhancer image-IT-Fx (Life Technologies, I36933) for 30 min at room temperature, and then blocked using BlockAid (Life Technologies, B10710) for 1 hr at room temperature. The FLAG-tagged NiV-F and mutants were detected by the anti-FLAG mouse monoclonal antibody (Sigma-Aldrich, F1804) and an Alexa Fluor 647 conjugated donkey anti-mouse secondary antibody (Invitrogen, A31571). Cells were incubated with primary antibody overnight at 4°C, and then with the secondary antibody for 1 hr at room temperature. Each antibody incubation is followed by five PBS washes, 5 min each time. Cells were then fixed in PBS containing 4% PFA for 10 min at room temperature.

## SMLM setup, imaging, and data analysis

SMLM setup. Imaging was performed on a custom-built SMLM. Briefly, the microscope was built upon an apochromatic TIRF oil-immersion objective lens (Nikon, 60×; numerical aperture 1.49). Four lasers were used for excitation: a 639-nm laser (MRL-FN-639, 500 mW) for exciting Alexa Fluor 647, a 532-nm laser (MGL-III-532-300 mW) for exciting cy3B, a 488-nm laser (MBL-F-473-300 mW) for exciting GFP, and a 405 nm laser (MDL-III-405-100 mW) for reactivating Alexa Fluor 647 and cy3B. For SMLM imaging, the exciting and reactivating lasers were combined into a single path using dichroic mirrors and then expanded and collimated using a beam expander. The incident light beam was focused on the back focal plane of the objective lens. A translation stage allowed the beam to be shifted between the center and edge of the objective lens so that the incident light emerging from the objective lens reaches the sample at various angles. For SMLM imaging of NiV-F at the cell and virus membranes at the cell-coverslip contact, the incident angle of the illumination light was adjusted to near or greater than the critical angle of the water–glass interface to allow total internal reflection and ensure low background fluorescence for single fluorophore detection (*Bates et al., 2013*). The emission fluorescence was separated using appropriate dichroic mirrors and filters (Semrock) and detected by electron-multiplying charge-coupled devices (Ixon, Andor). The full width at half maxima of the image of the point spread function was magnified in size to ~3 camera pixels (106 nm per pixel), ensuring the optimal fluorophore localization (*Bates et al., 2013*). A feedback loop, composed of a tracking camera (Andor Zyla SCOMS camera) and a computer-controlled piezo stage (Thorlabs 3-Axis NanoMax 300), was employed to control the sample drift to <1 nm laterally and 2.5 nm axially (*Tafteh et al., 2016.*)

Before SMLM imaging, fluorescence beads (Life technologies, F8799) were added to samples as fiducial markers for drift control. Samples were immersed in imaging buffer [TN buffer (50 mM Tris

(pH 8.0) and 10 mM NaCl), 0.5 mg/ml glucose oxidase, 40 µg/ml catalase, 10% glucose, and 50 mM mercaptoethylamine] (*Dempsey et al., 2011*). The expression level of the protein of interest in individual cells was determined by measuring the average emission fluorescence intensity of an area of $27 \times 27$ µm$^2$. To ensure sample stability during SMLM, a fiducial marker was continuously tracked by the tracking camera. A brief procedure for SMLM imaging is described as follows: (1) the sample was exposed to the imaging light beam [639 nm laser (1 kW/cm$^2$)] at a 90° incident angle to cause a subset of the fluorophores to switch to the fluorescent state; (2) the incident light angle was adjusted close to or at the critical angle at the water–glass interface to achieve total internal reflection; (3) a low density of activated fluorophores were recorded to avoid fluorophore overlap; (4) another subset of fluorophores was stochastically activated by exposure to the activation light (405 nm). This process was repeated until 40,000 images were acquired. Custom-written software in MATLAB (Mathworks) was used to identify activated fluorophores in each image, determine their precise locations by Gaussian fitting, and reconstruct SMLM images (*Bates et al., 2013*). Briefly, each image was first convolved with a Gaussian kernel to remove noise and background, and thresholded to search for local maxima. Next, two fitting steps were applied to determine the peak center position, the amplitude, and the width. Then, filtering and trail generation were combined to remove erroneous peaks and group multiple peaks originating from one activated fluorophore. Lastly, a list of fluorophore positions was reconstructed into an SMLM image. Clusters of NiV-F localizations on cell surface resolved by SMLM were identified and characterized using ClusDoC (*Pageon et al., 2016*). The min points and $\varepsilon$ were set at 4 and 20, respectively.

## VLP production and immunofluorescence

To produce NiV VLPs, HEK293T cells were transfected by NiV-F-FLAG or mutants, NiV-G-HA, and NiV-M-GFP at a 9:1:5 ratio by using polyethylenimine at 1 mg/ml (Polysciences, 23966-100). At 48 hr post-transfection, the supernatant of the cell culture was collected and subjected to ultracentrifugation on a 20% sucrose cushion at 125, 392 rcf for 90 min. The VLP-containing pellets were resuspended in 5% sucrose-NTE buffer. VLPs were bound to 2.5 µg fibronectin-coated 18 mm coverslips at 4°C overnight, followed by fixation using PBS containing 4% PFA and 0.2% glutaraldehyde. The protocols of immunofluorescence, SMLM setup, and SMLM imaging for cells were followed to stain and image NiV-F-FLAG and mutants on VLPs using SMLM. A widefield image of GFP was acquired and superimposed on the SMLM image of NiV-F. VLPs were identified as GFP-positive particles. The distribution of NiV-F localizations on VLPs was analyzed by using a custom-written OPTICS algorithm based on C++ and Point Cloud Library. Briefly, OPTICS segregates closely situated data points into clusters from a 3D SMLM dataset (*Ankerst et al., 1999*). OPTICS algorithm depends on two parameters: (1) $\varepsilon$ describes the radius for searching neighbors, and the number of neighbors of point $p$ is called $N_\varepsilon(p)$; (2) $minPts$ is a user-defined parameter; point $p$ is a core point if it has more than $minPts$ neighbors around it (including point $p$ itself). The core and reachability distances are calculated for cluster segregation. The core distance is the Euclidean distance of the $minPts$th closest points to point $p$ (*Equation 1*).

$$CoreDist_{\varepsilon,\, minPts}(p) = \begin{cases} UNDEFINED, \; if \; N_\varepsilon(p) < minPts, \\ minPts_{th} \; smallest \; distance \; in \; N_\varepsilon(p)\,, \; otherwise \end{cases} \tag{1}$$

The reachability distance of another point $t$ to point $p$ is either the Euclidean distance between point $t$ and point $p$ or the core distance of $p$, whichever is bigger (*Equation 2*).

$$ReachDist_{\varepsilon,\, minPts}(t,\, p) = \begin{cases} UNDEFINED, \; if \; N_\varepsilon(p) < minPts, \\ \max(CoreDist_{\varepsilon,\, minPts})(p)\,, EudDist(p,\, t)\,, \; otherwise \end{cases} \tag{2}$$

The process begins by randomly selecting a point (point A). If the number of neighbors of point A is larger than the user-defined minPts (set at 10), the algorithm outputs point A without a reachability-distance value. Next, OPTICS obtains point A's nearest neighbor (point B) and outputs the B–A reachability-distance value. Point B is the new starting point. OPTICS recalculates the reachability distance of all unprocessed neighbors of point B and selects the next point to process, as described

above. This process repeats until all points in the dataset have been visited. The final output of OPTICS is a sequence of reachability distances for each point.

The classification of the ordered sequence of reachability distances was performed by using custom-written 1D CNN based on Python and Tensorflow. Briefly, CNN extracts features from OPTICS output dataset by using convolutional layers for subsequent classification of the space information hidden in the OPTICS datasets. The convolutional layers use a 1D sliding window to slide along the sequence data (*Yu et al., 2014*). Each sliding window captures local information and passes it to the next layer. Multiple CNN layers work sequentially to build a high-dimensional representation of the dataset. A fully connected layer that transforms high-dimensional features into classification results is achieved by CNN convolution. We use t-SNE (t-distributed stochastic neighbor embedding) to reduce dimensionality on a trained neural network layer for visualization in *Figure 4C*; *Kobak and Berens, 2019*.

## Sodium dodecyl sulfate–polyacrylamide gel electrophoresis and western blot analysis and co-immunoprecipitation

For western blot analysis to confirm the expression of NiV-F-FLAG and mutants, HEK293T cells were seeded in 6-well plate and transfected with 2.5 µg pcDNA3.1 empty vector, NiV-F-FLAG, or mutants. At 28 or 48 hr posttransfection, cells were lysed in RIPA buffer (Millipore-Sigma, 20-188) supplemented with protease inhibitor (Sigma-Aldrich, 11836170001) on ice for 30 min. The cell lysates were collected after centrifuge at 16,000 × *g* for 20 min at 4°C. The cell lysates were supplemented with 1× sodium dodecyl sulfate (SDS) loading dye [60 mM Tris–HCl (pH = 6.8); 2% SDS; 10% glycerol, 0.025% Brophenol blue] and 15 mM DTT (dithiothreitol) (Thermo Scientific, R0861), and heated at 95°C for denature. The denatured cell lysates were loaded into 10% polyacrylamide gels for SDS–polyacrylamide gel electrophoresis (PAGE). Proteins were transferred to PVDF (polyvinylidene fluoride) membrane, pore size 0.45 µm (Cytiva, GE10600021). Membrane was blocked with 1% bovine serum albumin (Sigma-Aldrich, A9647-50G) in PBS and incubated with anti-FLAG mouse monoclonal antibody (Sigma-Aldrich, F1804). An HRP-conjugated goat anti-mouse secondary antibody and the clarity Western ECL substrate (Bio-Rad, 1705060) were used for protein detection. Images were acquired using Chemic Doc MP Imaging System (Bio-Rad).

For co-immunoprecipitation experiments, HEK293T cells were transfected with the following combinations (1) 2.3 µg empty pcDNA 3.1 vector and 0.2 µg AP2µ2-mcherry, (2) 0.2 µg AP2µ2-mcherry and 2.3 µg NiV-F-FLAG, and (3) 0.2 µg AP2µ2-mcherry and 2.3 µg NiV-F-YA. At 48 hrs post-transfection, cells from 1 well of a 6-well plate were washed with PBS and lysed in 200 µl lysis buffer provided with the µMACS DYKDDDDK isolation kit, and supplemented with protease inhibitors. Cells were isolated on ice for 30 min. Cell debris was removed by centrifuge at 16,000 × *g* for 20 min at 4°C. 60 µl of cell lysate was set aside for immunoblot analysis, and the rest was used for immunoprecipitation, as recommended by the manufacturer. 6 µl anti-DYKDDDDK microbeads (Miltenyi Biotec, 130-101-591) were added to 140 µl cell lysates and incubated for 30 min on ice. µ columns (Miltenyi Biotec, 130-101-591) were prepared according to the manufacturer's instructions. Lysate ran over the columns, and microbeads were washed according to the manufacturer's instruction. 20 µl preheated elution buffer (95°C) was added to the column before eluting the bound immunoprecipitated protein in 50 µl elution buffer. Elute was separated by 10% SDS–PAGE, and proteins were immunoblotted by mouse anti-FLAG and rabbit anti-mcherry primary antibodies. The HRP-conjugated goat anti-mouse and goat anti-rabbit secondary antibodies were used for protein detection.

### Flow cytometry

HEK293T cells were seeded in 6-well plate and transfected with 2.5 µg pcDNA3.1 empty vector, NiV-F-FLAG, or mutants. Cells were collected in 1 ml PBS containing 10 mM EDTA (Ethylenedi-aminetetraacetic acid) after 28 hr post-transfection. The collected cells were incubated on ice for 1 hr with primary antibody mouse anti-flag diluted in 1:200. Samples were washed twice in fluorescence-activated cell sorting buffer (0.1% fetal bovine serum with PBS). After washing, samples were incubated with fluorescent anti-mouse Alexa Fluor 647 antibody 1:400 on ice for 45 min. After being washed twice again, samples were read on a flow cytometer (Attune NxT Acoustic Focusing Cytometer, Thermo Fisher). The results were analyzed through FlowJo. The mean fluorescent intensities (MFIs) were normalized to the MFIs of the NiV F-WT.

## Cell–cell fusion

HEK293T cells were seeded in the 12-well plate and transfected with 1 µg total DNA of NiV-G and wt-F or mutant-F with a 3:1 ratio using Lipofectamine 3000. Cells were fixed with 4% PFA after 18 hr post-transfection. Four or more nuclei within a common cytoplasm were considered syncytium. Syncytia were quantified by counting the number of nuclei in the syncytium per 10 × filed (5 fields are counted per group) under microscope TE2000U.

## Viral entry kinetics assay

HEK293T cells were loaded with CCF2-AM in the loading buffer by using a CCF2-AM loading kit (Invitrogen) and incubated at room temperature for 90 min. Loaded cells were washed three times and resuspended in 60 µl of cold fusion buffer [25 mM HEPES (4-(2-Hydroxyethyl)piperazine-1-ethane-sulfonic acid)], 2 mM glutamine, 1 mM $CaCl_2$, 2.5 mM probenecid in HBSS (Hanks' balanced salt solution). 110 µl fusion buffer, 10 µl of VLPs, and 60 µl of loaded cells were added into a black, clear-bottom 96-well plate on ice. The VLPs were allowed to attach to target cells at 4°C for 1 hr. Viral entry was measured by a Tecan fluorometer preheated to 37°C. Fluorescence was recorded every 3 min for 4 hr. A blue-to-green (B/G) fluorescence ratio was determined for each well at each time point. The background control B/G ratio of VLPs expressing only MG and MF was then subtracted from each sample ratio as indicated, and the B/G ratio was plotted against time.

## VSV/NiV pseudovirus production, quantitative PCR, and viral entry quantification

HEK293T cells in a 10-cm dish were transfected with NiV-F-Flag and NiV-G-HA. pCDNA3.1 vector was transfected as the negative control. At 18–20 hr post-transfection, cells were infected by a recombinant VSV-ΔG-renilla luciferase virus in infection buffer (0.1% fetal bovine serum in PBS). After 48 hr, the supernatant was collected and concentrated by ultracentrifugation through a 20% sucrose cushion in NTE (NaCl-Tris-EDTA) buffer at 125,000 × $g$ at 4°C for 90 min. The supernatant was removed and the pellet was resuspended in the appropriate amount of 5% sucrose in NTE buffer. The expression levels of NiV-F and -G proteins in pseudoviruses were analyzed by western blot, and the VSV genome copy number was quantified by quantitative PCR (qPCR). Briefly, the viral RNA of NiV/VSV pseudoviruses was extracted by using a QIAGEN viral RNA extraction kit (QIAGEN, 52904), reverse transcribed to cDNA by the SuperScript III first-strand synthesis system (Invitrogen, 18080051). The VSV viral cDNA was quantified by using a qPCR Taqman probes (FAM-CGG TAT TTT TCC ATA ATT CAA GTA ATC TGC T-TAMRA) and the VSV-P plasmid was used as a standard to establish the standard curve. To quantify virus entry, Vero cells in a 96-well plate were infected using tenfold serial virus dilutions. Infections were done for 2 hr in infection buffer. After 24 hr, the luciferase activity was measured using a Renilla luciferase assay kit (Promega E2810). The relative light units (RLUs) were plotted against the viral genome copy number per milliliter in GraphPad Prism.

## Statistical analysis

Statistical analyses were performed using GraphPad Prism. The sample size, biological repeats, and statistical analysis used for each test were specified in the figure legend. The sample size was determined based on previous studies sharing similar objectives and methodology.

## Acknowledgements

We thank the multiscale imaging facility for instrument use. We thank Dr. Youssef Chebli at McGill University and Dr. David Scriven at the University of British Columbia for the support on laser scanning confocal microscopy and single-molecule imaging, respectively. Funding. This work is supported by funding from the Canadian Institutes of Health Research to Q.L. (183861) and the Coronavirus Variants Rapid Response Network to Q.L. (175622). Declare of interests. The authors declare no competing interests.

## Additional information

### Funding

| Funder | Grant reference number | Author |
| --- | --- | --- |
| Canadian Institutes of Health Research | 183861 | Qian Liu |
| Canadian Institutes of Health Research | 175622 | Qian Liu |

The funders had no role in study design, data collection and interpretation, or the decision to submit the work for publication.

### Author contributions

Qian Wang, Conceptualization, Data curation, Formal analysis, Validation, Visualization, Methodology, Writing – original draft, Writing – review and editing; Jinxin Liu, Data curation, Software, Formal analysis, Methodology, Writing – original draft, Writing – review and editing; Yuhang Luo, Jingjing Wang, Methodology, Writing – review and editing; Vicky Kliemke, Giuliana Leonarda Matta, Methodology; Qian Liu, Conceptualization, Resources, Data curation, Software, Formal analysis, Supervision, Funding acquisition, Validation, Investigation, Visualization, Methodology, Writing – original draft, Project administration, Writing – review and editing

### Author ORCIDs

Qian Liu https://orcid.org/0000-0002-6174-8982

Reviewer #1 (Public review): https://doi.org/10.7554/eLife.97017.3.sa1
Reviewer #2 (Public review): https://doi.org/10.7554/eLife.97017.3.sa2
Reviewer #3 (Public review): https://doi.org/10.7554/eLife.97017.3.sa3
Author response https://doi.org/10.7554/eLife.97017.3.sa4

## Additional files

### Supplementary files

MDAR checklist

### Data availability

All data generated or analyzed in this study are published with the article. Codes generated or used in this study are available at GitHub, copy archived at QLlab, 2024. Materials generated in this study are available from the corresponding author upon a Materials Transfer Agreement with McGill University.

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
