## [Editor Report · eLife Assessment]

This **valuable** study advances our understanding of how Nipah virus fusion protein F (NiV-F) organizes into nanoclusters on cell and viral membranes using biochemical and super-resolution microscopy methods. The conclusions are supported by **solid** evidence and the revision has addressed most of the reviewers' concerns. The relationship between clustering and fusion is of high interest and an interesting hypothesis to continue investigating in future studies.

---

## [Referee Report · Reviewer #1 (Public review)]

Summary:

In this work by Wang et al., the authors use single-molecule super-resolution microscopy together with biochemical assays to quantify the organization of Nipah virus fusion protein F (NiV-F) on cell and viral membranes. They find that these proteins form nanoscale clusters which favors membrane fusion activation, and that the physical parameters of these clusters are unaffected by protein expression level and endosomal cleavage. Furthermore, they find that the cluster organization is affected by mutations in the trimer interface on the NiV-F ectodomain and the putative oligomerization motif on the transmembrane domain, and that the clusters are stabilized by interactions among NiV-F, the AP2-complex, and the clathrin coat assembly. This work improves our understanding of the NiV fusion machinery, which may also have implications for our understanding of the function of other viruses.

Strengths:

The conclusions of this paper are well-supported by the presented data. This study sheds light on the activation mechanisms underlying the NiV fusion machinery.

---

## [Referee Report · Reviewer #2 (Public review)]

Summary:

In this manuscript, Wang and co-workets employ single molecule light microscopy (SMLM) to detect Nipah virus Fusion protein (NiV-F) in the surface of cells. They corroborate that these glycoproteins form microclusters (previously seen and characterized together with the NiV-G and Nipah Matrix protein by Liu and co-workers (2018) also with super-resolution light microscopy). Also seen by Liu and coworkers the authors show that the level of expression of NiV-F does not alter the identity of these microclusters nor endosomal cleavage. Moreover, mutations and the transmembrane domain or the hexamer-of-trimer interface seem to have a mild effect on the size of the clusters that the authors quantified. Importantly, it has also been shown that these particles tend to cluster in Nipah VLPs.

Strengths:

The authors have tried to perform SMLM in single VLPs and have shown partially the importance of NiV-F clustering.

Comments on the revised version:

I am happy with the answers the authors have provided to my questions

---

## [Referee Report · Reviewer #3 (Public review)]

Summary:

The manuscript by Wang and colleagues describes single molecule localization microscopy to quantify the distribution and organization of Nipah virus F expressed on cells and on virus-like particles. Notably the crystal structure of F indicated hexameric assemblies of F trimers. The authors propose that F clustering favors membrane fusion.

Strengths:

The manuscript provides solid data on imaging of F clustering with the main findings of:

- F clusters are independent of expression levels

- Proteolytic cleavage does not affect F clustering

- Mutations that have been reported to affect the hexamer interface reduce clustering on cells and its distribution on VLPs

- F nanoclusters are stabilized by AP

Comments on the revised version:

The authors addressed most of my previous concerns.

---

## [Author Response]

The following is the authors’ response to the original reviews.

**Public Reviews:**

**Reviewer #1 (Public Review):**
Summary:In this work by Wang et al., the authors use single-molecule super-resolution microscopy together with biochemical assays to quantify the organization of Nipah virus fusion protein F (NiV-F) on cell and viral membranes. They find that these proteins form nanoscale clusters which favors membrane fusion activation, and that the physical parameters of these clusters are unaffected by protein expression level and endosomal cleavage. Furthermore, they find that the cluster organization is affected by mutations in the trimer interface on the NiV-F ectodomain and the putative oligomerization motif on the transmembrane domain, and that the clusters are stabilized by interactions among NiV-F, the AP2-complex, and the clathrin coat assembly. This work improves our understanding of the NiV fusion machinery, which may have implications also for our understanding of the function of other viruses.Strengths:The conclusions of this paper are well-supported by the presented data. This study sheds light on the activation mechanisms underlying the NiV fusion machinery.Weaknesses:The authors provide limited details of the convolutional neural network they developed in this work. Even though custom-codes are made available, a description of the network and specifications of how it was used in this work would aid the readers in assessing its performance and applicability. The same holds for the custom-written OPTICS algorithm. Furthermore, limited details are provided for the imaging setup, oxygen scavenging buffer, and analysis for the single-molecule data, which limits reproducibility in other laboratories. The claim of 10 nm resolution is not backed up by data and seems low given the imaging conditions and fluorophores used. Fourier Ring Correlation analysis would have validated this claim. If the authors refer to localization precision rather than resolution, then this should be specified and appropriate data provided to support this claim.

We thank reviewer 1 for these suggestions. We described key steps in imaging setup, singlemolecule data reconstruction, the OPTICS algorithm in cluster identification, and 1D CNN in

classification of the OPTICS data in the Materials and Methods section. We also provided a recipe for the imaging buffer. We refer to 10 nm localization precision rather than resolution. The localization precision achieved by our SMLM system is shown in the Author response image 1.

**Author response image 1. sa4fig1:** The localization precision of the custom-built SMLM. Shows the distribution of localization error at the x (dX), y (dY), and z (dZ) direction in nanometer of blinks generated from Alexa Flour 647 labeled to NiV-F expressed on the plasma membrane of PK13 cells. The lateral precision is <10 nm and the axial precision is < 20 nm.

**Reviewer #2 (Public Review):**
Summary:In this manuscript, Wang and co-workers employ single molecule light microscopy (SMLM) to detect NiV fusion protein (NiV-F) in the surface of cells. They corroborate that these glycoproteins form microclusters (previously seen and characterized together with the NiVG and Nipah Matrix protein by Liu and co-workers (2018) also with super-resolution light microscopy). Also seen by Liu and coworkers the authors show that the level of expression of NiV-F does not alter the identity of these microclusters nor endosomal cleavage. Moreover, mutations and the transmembrane domain or the hexamer-of-trimer interface seem to have a mild effect on the size of the clusters that the authors quantified.Importantly, it has also been shown that these particles tend to cluster in Nipah VLPs.

We thank reviewer #2 for the comments and suggestions. This paper is built on Liu et al 1 to further characterize the nanoclusters formed by NiV-F and their role in membrane fusion activation. While Liu et al. studied the NiV glycoprotein distribution at the NiV assembly sites to inform mechanisms in NiV assembly and release, Wang et al. analyzed the nanoorganization and distribution of NiV-F at the prefusion conformation, providing insights into the membrane fusion activation mechanisms.

Strengths:The authors have tried to perform SMLM in single VLPs and have shown partially the importance of NiV-F clustering.Weaknesses:The labelling strategy for the NiV-F is not sufficiently explained. The use of a FLAG tag in the extracellular domain should be validated and compared with the unlabelled WT NiV-F when expressed in functional pseudoviruses (for example HIV-1 based particles decorated with NiV-F). This experiment should also be carried out for both infection and fusion (including BlaM-Vpr as a readout for fusion). I would also suggest to run a time-of-addition BlaM experiment to understand how this particular labelling strategy affects single virion fusion as compared to the the WT.

We thank reviewer #2 for this suggestion. We have made various efforts to validate the expression and function of FLAG-tagged NiV-F. The NiV-F-FLAG shows comparable cell surface expression levels and induces similar cell-cell fusion levels in 293T cells as that of untagged NiV-F 1. The NiV-F-FLAG also showed similar levels of virus entry as untagged NiV-F when both were pseudotyped on a recombinant Vesicular Stomatitis Virus (VSV) with the VSV glycoprotein replaced by a *Renilla* luciferase reporter gene (VSV-ΔG-rLuc; Fig. S1D). We also performed a virus entry kinetics assay using NiV VLPs expressing NiV-M-βlactamase (NiV-M-Bla), NiV-G-HA, and NiV-F-FLAG, NiV-F-AU1 or untagged NiV-F. The intracellular AU1 tag is located at the C-terminus of NiV-F (Genbank accession no. AY816748.1). However, we detected different levels of NiV-M-Bla in equal volume of VLPs, suggesting that the tags in NiV-F affect the budding of the VLPs (Author response image 2A). Therefore, we performed fusion kinetics assay by using VLPs expressing the same levels of NiV-M-Bla. Among them, the NiV-F-FLAG on VLPs shows the most efficient fusion between VLP and HEK293T cell membranes (Author response image 2B), significantly more efficient than that of untagged NiV-F and NiV-FAU1. However, we cannot attribute the enhanced fusion activity to the FLAG tag, because the readout of this assay relies on both the levels of β-lactamase (introduced by NiV-M-Bla in VLPs) and the NiV-F constructs. The tags in NiV-F could affect both the budding of VLPs and the stoichiometry of F and M in individual VLPs. We did not use the HIV-based pseudovirus system because the incorporation of NiV-F into HIV pseudoviruses requires a C-terminal deletion 2,3.

In summary, the FLAG tag does not affect cell-cell fusion 1 and virus entry when pseudotyped to the recombinant VSV-ΔG-rLuc viruses (Fig. S1D). Given that we do not observe any difference in clustering between an HA- and FLAG-tagged NiV-F constructs on PK13 cell surface (Fig. S1A-C), we conclude that the FLAG tag has minimal effect on both the fusion activity and the nanoscale distribution of NiV-F.

**Author response image 2. sa4fig2:** Viral entry is not affected by labeling of NiV-F. (A) Western blot analysis of NiV-M-Bla in NiV-VLPs generated by HEK293T cells expressing NiV-M-Bla, NiV-G-HA and NiV-F-FLAG, untagged NiV-F, or NiV-F-AU1. Equal volume of VLPs were separated by a denaturing 10% SDS–PAGE and probed against β-lactamase (SANTA CRUZ, sc-66062). (B) NiV-VLPs expressing NiV-M-BLa, NiV-G-HA, and NiV-F-FLAG, untagged NiV-F or NiV-F-AU1 expression plasmids were bond to the target HEK293T cells loaded with CCF2-AM dye at 4°C. The Blue/Green (B/G) ratio was measured at 37°C for 4 hrs at a 3-min interval. Results were normalized to the maximal B/G ratio of NiV-F-FLAG-NiV VLPs. Results from one representative experiment out of three independent experiments are shown.

It would also be very important to compare the FLAG labelling approach with recent advances in the field (for instance incorporating noncanonical amino acids (ncAAs) into NiVF by amber stop-codon suppression, followed by click chemistry).

We are greatly thankful for this comment from reviewer #2. Labeling noncanonical amino acids (ncAAs) with biorthogonal click chemistry is indeed a more precise labeling strategy compared to the traditional epitope labeling approach used in this paper. We will explore the applications of ncAAs labeling in single-molecule localization imaging and virus-host interactions in future projects.

In this paper, the FLAG tag inserted in NiV-F protein seems to have minimal effect on the NiV-F-induced virus entry and cell-cell fusion 1 (Fig. S1). Although the FLAG tag labeling approach may increase the detectable size of NiV-F nanoclusters due to the use of the antibody complex, it should not affect our conclusions drawn from the relative comparisons between wt and mutant NiV-F or control and drug-treated cells.

The correlation between the existence of microclusters of a particular size and their functionality is missing. Only cell-cell fusion assays are shown in supplementary figures and clearly, single virus entry and fusion cannot be compared with the biophysics of cell-cell fusion. Not only the environment is completely different, membrane curvature and the number of NiV-F drastically varies also. Therefore, specific fusion assays (either single virus tracking and/or time-of-addition BlaM kinetics with functional pseudoviruses) are needed to substantiate this claim.

We thank Reviewer 2 for the suggestion. To support the link between F clustering and viruscell membrane fusion, we conducted pseudotyped virus entry and VLP fusion kinetics assays, as shown in revised Figure S4. The viral entry results (Fig. S4 E and F) corroborate that of the cell-cell fusion assay (Fig. S4A and B) and previously published data 4. The fusion kinetics confirmed that the real-time fusion kinetics was affected by mutations at the hexameric interface, with the hypo-fusogenic mutants L53D and V108D exhibited reduced entry efficiency while the hyper-fusogenic mutant Q393L showed increased efficiency (Fig. S4G and H). The results were described in detail in the revised manuscript.

Additionally, we performed a pseudotyped virus entry assay on the LI4A (Fig. S6F and G) and YA (Fig. S7F and G) mutants to verify the function of these mutants on viruses in revised Supplemental Figures. Neither LI4A nor YA incorporated into the VSV/NiV pseudotyped viruses as shown by the Western blot analyses of the pseudovirions (Fig. S6F and S7F), and thus did not induce virus entry, consisting with the cell-cell fusion results (Fig. S6C, D and Fig. S7C, D). We did not perform the entry kinetic assay of these two mutants as they do not incorporate into VLPs or pseudovirions.

The authors also claim they could not characterize the number of NiV-F particles per cluster. Another technique such as number and brightness (Digman et al., 2008) could support current SMLM data and identify the number of single molecules per cluster. Also, this technology does not require complex microscopy apparatus. I suggest they perform either confocal fluorescence fluctuation spectroscopy or TIRF-based nandb to validate the clusters and identify how many molecule are present in these clusters.

We thank reviewer 2 for this suggestion. Determining the true copy number of NiV-F in individual clusters could verify whether the F clusters on the plasma membrane are hexamer-of-trimer assemblies. Regardless, it does not affect our conclusion that the organization of NiV-F into nanoclusters affects the membrane fusion triggering ability. The confocal fluorescence fluctuation spectroscopy (FFS) and TIRF-based analyses are accessible tools for quantifying fluorophore copy numbers and/or stoichiometry based on fluorescence fluctuation or photobleaching. However, these methods are unable to quantify the number of proteins in individual clusters because they analyze fluorophores either in the entire cell (as in wide-field epifluorescence microscopy coupled with FFS and TIRF-coupled photobleaching) 5–7 or within a large excitation volume (confocal laser scanning microscopycoupled FFS) 8. Both of these volumes are significantly larger than a single NiV-F cluster, which has an average diameter of 24-26 nm (Fig. 1F).

The current SMLM setup is useful for characterizing the protein distribution and organization. However, quantifying the true protein copy number within a nanocluster is challenging because of the stochasticity of fluorophore blinking and the unknown labeling stoichiometry 9–11. To address the challenge in fluorophore blinking, quantitative DNA-PAINT (qDNA-PAINT) may be used because the on-off frequency of the fluorophores is tied to the well-defined kinetic constants of DNA binding and the influx rate of the imager strands, rather than the stochasticity of fluorophore blinking. Thus, the frequency of blinks can be translated to protein counting 12. To address the challenge in unknown labeling stoichiometry, DNA origami can be used as a calibration standard 11. DNA origami supports handles at a regular space with several to tens of nanometers apart, and the handles can be conjugated with a certain number of proteins of interest. The copy number of protein interest in the experimental group can be determined by comparing the SMLM localization distribution of the sample to that of the DNA origami calibration standard. Given the requirement of a more sophisticated SMLM setup and a high-precision calibration tool, we will explore the quantification of NiV-F copy numbers in nanoclusters in a future project.

Also, it is not clear how many cells the authors employ for their statistics at least 30-50 cells should be employed and not consider the number of events blinking events. I hope the authors are not considering only a single cell to run their stats... The differences between the mutants and the NiV-F is minor even if their statistical analyses give a difference (they should average the number and size of the clusters per cell for a total of 30-50 cells with experiments performed at least in three different cells following the same protocol). Overall, it seems that the authors have only evaluated a very low number of cells.

We disagree with this comment from Reviewer #2. The sample size for cluster analysis in SMLM images was chosen by considering the target of the study (cells and VLPs) and the data acquisition and analysis standards in the SMLM imaging field. We also noted the sample size (# of ROI and cells) in the figure legend.

Below, we compared the sample sizes in our study to those in similar studies that used comparable imaging and cluster analysis methods from 2015 to 2024. The classical clustering analysis methods are categorized into global clustering (*e.g.* nearest neighbor analysis, Ripley’s K function, and pair correlation function) and complete clustering, such as density-based analysis (*e.g.* DBSCAN, Superstructure, FOCAL, ToMATo) and Tessellationbased analysis (*e.g.* Delaunay triangulation, Voronoii Tessellation). The global clustering analysis method provides spatial statistics for global protein clustering or organization (*e.g.* clustering extent), while the complete clustering approach extracts information from a single-cluster level, such as the morphology and localization density of individual clusters. We used the density-based analyses, DBSCAN and OPTICS, for cluster analysis on cell plasma membranes and VLP membranes.

**Author response table 1. sa4table1:** The comparison of imaging methods, analysis methods, and sample size in the current study to other studies conducted from 2015 to 2024.

Study	Acquisition	Analysis
Ref.	Year	Target		Methods	Sample size
Wang et al. (this study)	2024	NiV fusion protein in PK13, HeLa cells	dSTORM	DBSCAN, Hopkins	4–20 cells, ROI 40–242 per condition from 3 independent experiments.
Wang et al. (this study)	2024	NiV fusion protein in VLPs	dSTORM	OPTICS	40–200 VLPs from three independent experiments
Rubin-Delancy et al.^13^	2015	CD3 in T cells	PALM, dSTORM	Ripley, Bayesian	30 ROIs per condition.
Griffe et al.^14^	2017	LAT vesicles in T cells	iPALM	Ripley, Bayesian	5 cells per condition
Griffie et al.^15^	2017	CD4 in T cells	Live-cell PALM		6 cells
Caetano et al.^16^	2015	PACS, LAMP1 in HeLa cells	Ground State Depletion Microscopy	Density-based, Ripley	5 cells per experiment (3 exp.)
Malkusch and Heilemann^17^	2016	HIV, gag, env in T cells	SMLM	DBSCAN, Ripley, OPTICS	1 cell
Zhang et al.^18^	2017	*Salmonella* typhimurium mutants in bacterial cells	FPALM	DBSCAN	58–60 bacterial cells.
Tobin et al.^19^	2018	HER2 receptor in breast cancer cells	dSTORM	Density pair correlation	17–23 cells
Levet et al.^20^	2015	Microtubules in Cos-7 cells, GluA1, tubulin, integrin-β3 in neuron cells	PALM, dSTORM	Voronoi	3 cells per condition
Peters et al.^21^	2018	F-actin in T cells, microtubule network in HeLa cells	dSTORM	Voronoi, Ripley	3–5 cells
Levet et al.^22^	2019	Nuclear pore complex; microtubule, and actin cytoskeleton regulators	DNA-PAINT, dSTORM, and PALM	Voronoi	3–18 cells per condition.
Banerjee et al.^23^	2023	ULK in autophagy formation in HeLa cells	PALM	Spatial cross correlation	7–34 cells per condition
Pageon et al.^24^	2016	CD3ζ–PSCFP2 in Jurkat–OT-I cells	PALM	DBSCAN	4–6 cells
Cresens et al.^25^	2023	Flat clathrin lattces in KM12L4a cell	PALM	DBSCAN	35 cells
Seeling et al.^26^	2023	Decγn-1 in FγRIIb cell	dSTORM	DBSCAN	6–10 cells

They should also compare the level of expression (with the number of molecules per cell provided by number and brightness) with the total number of clusters.

We thank reviewer 2 for this suggestion. We compared the level of expression with the total number of clusters for F-WT in Figure 1I in the main text.

The same applies to the VLP assay. I assume the authors have only taken VLPs expressing both NiV-M and NiV-F (and NiV-G). But even if this is not clearly stated I would urge the authors to show how many viruses were compared per condition normally I would expect 300 particles per condition coming from three independent experiments. As a negative control to evaluate the cluster effect I would mix the different conditions. Clearly you have clusters with all conditions and the differences in clustering depending on each condition are minimal. Therefore you need to increase the n for all experiments.

We thank reviewer 2 for this comment. We acquired and analyzed more images of NiV VLPs bearing F-WT, Q393L, L53D, and V108D. Results are shown in the revised Figure 4 and the number of VLPs (>300) used for analysis is specified in the figure legend. An increased number of VLP images does not affect the classification result in Figure 4C.

As for the suggestion on “evaluating the cluster effect at different mixed conditions”, I assume that reviewer 2 would like to see how the presence of different viral structural proteins (F, M, and G) on VLPs could affect F clustering. We showed that the organization of NiV envelope proteins on the VLP membrane is similar in the presence or absence of NiV-M by direct visualization 27, suggesting that the effect of NiV-M on F-WT clustering on VLPs is minimal. We also show comparable incorporation of NiV-F among the NiV-F hexamer-oftrimer mutants (Fig. 4A). Therefore, we did not test the F clustering at different F, M, and G combinations in this paper. However, this could be an interesting question to pursue in a paper focusing on NiV VLP production.

**Reviewer #3 (Public Review):**
Summary:The manuscript by Wang and colleagues describes single molecule localization microscopy to quantify the distribution and organization of Nipah virus F expressed on cells and on virus-like particles. Notably the crystal structure of F indicated hexameric assemblies of F trimers. The authors propose that F clustering favors membrane fusion.Strengths:The manuscript provides solid data on imaging of F clustering with the main findings of:- F clusters are independent of expression levels- Proteolytic cleavage does not affect F clustering- Mutations that have been reported to affect the hexamer interface reduce clustering on cells and its distribution on VLPs - - F nanoclusters are stabilized by APWeaknesses:The relationship between F clustering and fusion is per se interesting, but looking at F clusters on the plasma membrane does not exclude that F clustering occurs for budding. Many viral glycoproteins cluster at the plasma membrane to generate micro domains for budding.This does not exclude that these clusters include hexamer assemblies or clustering requires hexamer assemblies.

We thank reviewer #3 for this question. We did not focus on the role of NiV-F clusters for budding in the current manuscript, although this is an interesting topic to pursue. In this manuscript, we observed that NiV VLP budding is decreased for some cluster-disrupting mutants, such as F-YA, and F-LI4A. however, F-V108D showed increased budding compared to F-WT (Fig. 4A). We also observed that VLPs and VSV/NiV pseudoviruses expressing L53D have little NiV-G (Fig. 4A, Fig. S4F and S4H), although the incorporation level of L53D is comparable to that of wt F in both VLPs and pseudovirions (Fig. 4A and Fig. S4F). L53D is a hypofusogenic mutant with decreased clustering ability. Therefore, our current data do not show a clear link between F clustering and NiV VLP budding or glycoprotein incorporation.

We reported that both NiV-F and -M form clusters at the plasma membrane although NiV-F clusters are not enriched at the NiV-M positive membrane domains 1. This result indicates that NiV-M is the major driving force for assembly and budding, while NiV-F is passively incorporated into the assembly sites. The central role of NiV-M in budding is also supported by a recent study showing that NiV-M induces membrane curvature by binding to PI(4,5)P2 in the inner leaflet of the plasma membrane 28. However, the expression of NiV-F alone induces the production of vesicles bearing NiV-F 29 and NiV-F recruits vesicular trafficking and actin cytoskeleton factors to VLPs either alone or in combination with NiV-G and -M, indicating a potential autonomous role in budding 30. Additionally, several electron microscopy studies show that the paramyxovirus F forms 2D lattice interspersed above the M lattice, suggesting the participation of F in virus assembly and budding. Nonetheless, the evidence above suggests that NiV-F may play a role in budding, but our data cannot correlate NiV-F clustering to budding.

Assuming that the clusters are important for entry, hexameric clusters are not unique to Nipah virus F. Similar hexameric clusters have been described for the HEF on influenza virus C particles (Halldorsson et al 2021) and env organization on Foamy virus particles (Effantin et al 2016), both with specific interactions between trimers. What is the organization of F on Nipah virus particles? If F requires to be hexameric for entry, this should be easily imaged by EM on infectious or inactivated virus particles.

We thank reviewer #3 for this suggestion. The hexamer-of-trimer NiV-F is observed on the VLP surface by electron tomography 4. The NiV-F hexamer-of-trimers are arranged into a soccer ball-like structure, with one trimer being part of multiple hexamer-of-trimers. The implication of NiV-F clusters in virus entry and the potential mechanism for NiV-F higherorder structure formation are discussed in the revised manuscripts.

AP stabilization of the F clusters is curious if the clusters are solely required for entry? Virus entry does not recruit the clathrin machinery. Is it possible that F clusters are endocytosed in the absence of budding?

We thank reviewer #3 for this question. The evidence from the current study does not exclude the role of NiV-F clustering in virus budding. NiV-F is known to be endocytosed in the virus-producing cells for cleavage by Cathepsin B or L at endocytic compartments at a pH-dependent manner31–33 in the absence of budding. However, given that all cleaved and uncleaved NiV-F have an endocytosis signal sequence at the cytoplasmic tail and are able to interact with AP-2 for endosome assembly and the cleaved and uncleaved F may have similar clustering patterns (Fig. 2), we do not think NiV-F clustering is specifically regulated for the cleavage of NiV-F. A plausible hypothesis is that NiV-F clusters are stabilized by multiple intrinsic factors (e.g. trimer interface) and host factors (e.g. AP-2) on cell membrane for cell-cell fusion and virus budding. We linked the clustering to the fusion ability of NiV-F in this study, but the NiV-F clustering may also be important in facilitating virus budding. Once in the viruses, the higher-order assembly of the clusters (*e.g.* lattice) may form due to protein enrichment, and the cell factors may not be the major maintenance force.

Clusters are required for budding.Other points:Fig. 3: Some of the V108D and L53D clusters look similar in size than wt clusters. It seems that the interaction is important but not absolutely essential. Would a double mutant abrogate clustering completely?

We thank Reviewer #3 for the suggestion. We generated a double mutant of NIV-F with L53D and V108D (NiV-F-LV) and assessed its expression and processing. Although the mutant retained processing capability, it exhibited minimal surface expression, making it unfeasible to analyze its nano-organization on the cell or viral membrane.

**Author response image 3. sa4fig3:** The expression and fusion activity of Flag-tagged NiV-F and NiV-F L53D-V108D (LV). (A) Representative western blot analysis of NiV-F-WT, LV in the cell lysate of 293T cells. 293T cells were transfected by NiV-F-WT or the LV mutant. The empty vector was used as a negative control. The cell lysates were analyzed on SDS-PAGE followed by western blotting after 28hrs post-transfection. F0 and F2 were probed by the M2 monoclonal mouse antiFLAG antibody. GAPDH was probed by monoclonal mouse anti-GAPDH. (B) Representative images of 293T cell-cell fusion induced by NiV-G and NiV-F-WT or NiV-F-LV. 293T cells were co-transfected with plasmids coding for NiV-G and empty vector (NC) or NiV-F constructs. Cells were fixed at 18 hrs post-transfection. Arrows point to syncytia. Scale bar: 10um. (C) Relative cell-cell fusion levels in 293T cells in (B). Five fields per experiment were counted from three independent experiments. Data are presented as mean ± SEM. (D) The cell surface expression levels of NiV-F-WT, NiV-F-LV in 293T cells measured by flow cytometry. Mean fluorescence Intensity (MFI) values were calculated by FlowJo and normalized to that of F-WT. Data are presented as mean ± SEM of three independent experiments. Statistical significance was determined by the unpaired t-test with Welch’s correction (*P<0.05, **P<0.01, ***P<0.001, ****P<0.0001). Values were compared to that of the NiV-F-WT.

Fig. 4: The distribution of F on VLPs should be confirmed by cryoEM analyses. This would also confirm the symmetry of the clusters. The manuscript by Chernomordik et al. JBC 2004 showed that influenza HA outside the direct contact zone affects fusion, which could be further elaborated in the context of F clusters and the fusion mechanism.

We thank reviewer 3 for this suggestion. The distribution of F on VLPs was resolved by electron tomogram which showed that the NiV-F hexamer-of-trimers are arranged into a soccer ball-like structure 4. The role of influenza HA outside of the contact zone in fusion activation is an interesting phenomenon. It may address the energy transmission within and among clusters. We will pursue this topic in a future project.

**Recommendations for the authors:**

**Reviewer #1 (Recommendations For The Authors):**
• Please define all used abbreviations throughout the manuscript and in the SI.

We defined the abbreviations at their first usage.

• The sentence starting with "Additionally, ..." on line 155 appears to be incomplete.

We corrected this sentence.

• The statement starting with "As reported, ..." on line 181 should be supported by a reference.

We added a reference.

• In Fig. 4C, it is unclear what the x and y axes represent.

Fig. 4C is a t-SNE plot for visualizing high-dimensional data in a low-dimensional space. It maintains the local data structure but does not represent exact quantitative relationships. In other words, points that are close together in Fig. 4C are also close in the high-dimensional space, meaning the OPTICS plots, which reflect the clustering patterns, are similar for two points that are positioned near each other in Fig. 4C. Therefore, the x and y axes do not represent the original, quantitative data, and thus the axis titles are meaningless.

• The reference on line 306 appears to be unformatted.

We reformatted the reference.

**Reviewer #2 (Recommendations For The Authors):**
The authors need to include the overall statistics for each experiment (at least 30 to 50 cells with three independent experiments are needed).

We highlighted the sample size (number of ROI and number of cells) used for analysis in the figure legend. The determination of the sample size is justified in Table 1 in the response letter.

The authors need to generate a functional pseudovirus system (for example HIVpp/NiV F) to run both infectivity and fusion experiments (including Apr-BlaM assay).

We tested viral entry using a VSV/NiV pseudovirus system and the viral entry kinetics using VLPs expressing NiV-M-β-lactamase. The results are presented in Fig. S1, S4, S6, and S7.

**Reviewer #3 (Recommendations For The Authors):**
Even low resolution EM data on VLPs or viruses would strengthen the conclusions.

We thank this reviewer for the suggestion. We cited the NiV VLP images acquired by electron tomography 4, but we currently have limited resources to perform cryoEM on NiV VLPs.

References.

(1) Liu, Q., Chen, L., Aguilar, H. C. & Chou, K. C. A stochastic assembly model for Nipah virus revealed by super-resolution microscopy. *Nature Communications* 9, 3050 (2018).

(2) Khetawat, D. & Broder, C. C. A Functional Henipavirus Envelope Glycoprotein Pseudotyped Lentivirus Assay System. *Virology Journal* 7, 312 (2010).

(3) Palomares, K. *et al.* Nipah Virus Envelope-Pseudotyped Lentiviruses Efficiently Target ephrinB2Positive Stem Cell Populations In Vitro and Bypass the Liver Sink When Administered In Vivo. *J Virol* 87, 2094–2108 (2013).

(4) Xu, K. *et al.* Crystal Structure of the Pre-fusion Nipah Virus Fusion Glycoprotein Reveals a Novel Hexamer-of-Trimers Assembly. *PLoS Pathog* 11, e1005322 (2015).

(5) Bakker, E. & Swain, P. S. Estimating numbers of intracellular molecules through analysing fluctuations in photobleaching. *Sci Rep* 9, 15238 (2019).

(6) Nayak, C. R. & Rutenberg, A. D. Quantification of Fluorophore Copy Number from Intrinsic

Fluctuations during Fluorescence Photobleaching. *Biophys J* 101, 2284–2293 (2011).

(7) Salavessa, L. & Sauvonnet, N. Stoichiometry of ReceptorsReceptors at the Plasma MembranePlasma membrane During Their EndocytosisEndocytosis Using Total Internal Reflection Fluorescent (TIRF) MicroscopyMicroscopy Live Imaging and Single-Molecule Tracking. in *Exocytosis and Endocytosis: Methods and Protocols* (eds. Niedergang, F., Vitale, N. & Gasman, S.) 3–17 (Springer US, New York, NY, 2021). doi:10.1007/978-1-0716-1044-2_1.

(8) Slenders, E. *et al.* Confocal-based fluorescence fluctuation spectroscopy with a SPAD array detector. *Light Sci Appl* 10, 31 (2021).

(9) Annibale, P., Vanni, S., Scarselli, M., Rothlisberger, U. & Radenovic, A. Identification of clustering artifacts in photoactivated localization microscopy. *Nat Methods* 8, 527–528 (2011).

(10) Baumgart, F. *et al.* Varying label density allows artifact-free analysis of membrane-protein nanoclusters. *Nat Methods* 13, 661–664 (2016).

(11) Zanacchi, F. C. *et al.* A DNA origami platform for quantifying protein copy number in super-resolution. *Nat Methods* 14, 789–792 (2017).

(12) Jungmann, R. *et al.* Multiplexed 3D cellular super-resolution imaging with DNA-PAINT and Exchange-PAINT. *Nature Methods* 11, 313–318 (2014).

(13) Rubin-Delanchy, P. *et al.* Bayesian cluster identification in single-molecule localization microscopy data. *Nat Methods* 12, 1072–1076 (2015).

(14) Griffié, J. *et al.* 3D Bayesian cluster analysis of super-resolution data reveals LAT recruitment to the T cell synapse. *Sci Rep* 7, 4077 (2017).

(15) Dynamic Bayesian Cluster Analysis of Live-Cell Single Molecule Localization Microscopy Datasets - Griffié - 2018 - Small Methods - Wiley Online Library. doi.org/10.1002/smtd.201800008.

(16) Caetano, F. A. *et al.* MIiSR: Molecular Interactions in Super-Resolution Imaging Enables the Analysis of Protein Interactions, Dynamics and Formation of Multi-protein Structures. *PLOS Computational Biology* 11, e1004634 (2015).

(17) Malkusch, S. & Heilemann, M. Extracting quantitative information from single-molecule superresolution imaging data with LAMA – LocAlization Microscopy Analyzer. *Sci Rep* 6, 34486 (2016).

(18) Zhang, Y., Lara-Tejero, M., Bewersdorf, J. & Galán, J. E. Visualization and characterization of individual type III protein secretion machines in live bacteria. *Proceedings of the National Academy of Sciences* 114, 6098–6103 (2017).

(19) Tobin, S. J. *et al.* Single molecule localization microscopy coupled with touch preparation for the quantification of trastuzumab-bound HER2. *Sci Rep* 8, 15154 (2018).

(20) Levet, F. *et al.* SR-Tesseler: a method to segment and quantify localization-based super-resolution microscopy data. *Nature Methods* 12, 1065–1071 (2015).

(21) Peters, R., Griffié, J., Burn, G. L., Williamson, D. J. & Owen, D. M. Quantitative fibre analysis of singlemolecule localization microscopy data. *Sci Rep* 8, 10418 (2018).

(22) Levet, F. *et al.* A tessellation-based colocalization analysis approach for single-molecule localization microscopy. *Nat Commun* 10, (2019).

(23) Banerjee, C. *et al.* ULK1 forms distinct oligomeric states and nanoscopic structures during autophagy initiation. *Science Advances* 9, eadh4094 (2023).

(24) Pageon, S. V. *et al.* Functional role of T-cell receptor nanoclusters in signal initiation and antigen discrimination. *Proceedings of the National Academy of Sciences* 113, E5454–E5463 (2016).

(25) Cresens, C. *et al.* Flat clathrin lattices are linked to metastatic potential in colorectal cancer. *iScience* 26, 107327 (2023).

(26) Seeling, M. *et al.* Immunoglobulin G-dependent inhibition of inflammatory bone remodeling requires pattern recognition receptor Dectin-1. *Immunity* 56, 1046-1063.e7 (2023).

(27) Liu, Q. T. *et al.* The nanoscale organization of Nipah virus matrix protein revealed by super-resolution microscopy. *Biophysical Journal* 121, 2290–2296 (2022).

(28) Norris, M. J. *et al.* Measles and Nipah virus assembly: Specific lipid binding drives matrix polymerization. *Science Advances* 8, eabn1440 (2022).

(29) Patch, J. R. *et al.* The YPLGVG sequence of the Nipah virus matrix protein is required for budding. *Virol. J.* 5, 137 (2008).

(30) Johnston, G. P. *et al.* Nipah Virus-Like Particle Egress Is Modulated by Cytoskeletal and Vesicular Trafficking Pathways: a Validated Particle Proteomics Analysis. *mSystems* 4, e00194-19 (2019).

(31) Diederich, S. *et al.* Activation of the Nipah Virus Fusion Protein in MDCK Cells Is Mediated by Cathepsin B within the Endosome-Recycling Compartment. *J Virol* 86, 3736–3745 (2012).

(32) Diederich, S., Thiel, L. & Maisner, A. Role of endocytosis and cathepsin-mediated activation in Nipah virus entry. *Virology* 375, 391–400 (2008).

(33) Pager, C. T., Craft, W. W., Patch, J. & Dutch, R. E. A mature and fusogenic form of the Nipah virus fusion protein requires proteolytic processing by cathepsin L. *Virology* 346, 251–257 (2006).